# AKAP5 complex facilitates purinergic modulation of vascular L-type Ca$^{2+}$ channel Ca$_V$1.2

Maria Paz Prada[1,8], Arsalan U. Syed[1,8], Gopireddy R. Reddy[1,8], Miguel Martín-Aragón Baudel [1], Víctor A. Flores-Tamez [1], Kent C. Sasse[2], Sean M. Ward[3], Padmini Sirish[4], Nipavan Chiamvimonvat[1,4,5], Peter Bartels [1], Eamonn J. Dickson[6], Johannes W. Hell [1], John D. Scott[7], Luis F. Santana[6], Yang K. Xiang [1,5], Manuel F. Navedo [1✉] & Madeline Nieves-Cintrón [1✉]

The L-type Ca$^{2+}$ channel Ca$_V$1.2 is essential for arterial myocyte excitability, gene expression and contraction. Elevations in extracellular glucose (hyperglycemia) potentiate vascular L-type Ca$^{2+}$ channel via PKA, but the underlying mechanisms are unclear. Here, we find that cAMP synthesis in response to elevated glucose and the selective P2Y$_{11}$ agonist NF546 is blocked by disruption of A-kinase anchoring protein 5 (AKAP5) function in arterial myocytes. Glucose and NF546-induced potentiation of L-type Ca$^{2+}$ channels, vasoconstriction and decreased blood flow are prevented in AKAP5 null arterial myocytes/arteries. These responses are nucleated via the AKAP5-dependent clustering of P2Y$_{11}$/ P2Y$_{11}$-like receptors, AC5, PKA and Ca$_V$1.2 into nanocomplexes at the plasma membrane of human and mouse arterial myocytes. Hence, data reveal an AKAP5 signaling module that regulates L-type Ca$^{2+}$ channel activity and vascular reactivity upon elevated glucose. This AKAP5-anchored nanocomplex may contribute to vascular complications during diabetic hyperglycemia.

---

[1] Department of Pharmacology, University of California Davis, Davis, CA 95616, USA. [2] Sasse Surgical Associates, Reno, NV 89502, USA. [3] Department of Physiology and Cell Biology, University of Nevada Reno, Reno, NV 89557, USA. [4] Department of Internal Medicine, University of California Davis, Davis, CA 95616, USA. [5] VA Northern California Healthcare System, Mather, CA 95655, USA. [6] Department of Physiology and Membrane Biology, University of California Davis, Davis, CA 95616, USA. [7] Department of Pharmacology, University of Washington Seattle, Seattle, WA 98195, USA. [8]These authors contributed equally: Maria Paz Prada, Arsalan U. Syed, Gopireddy R. Reddy. ✉email: mfnavedo@ucdavis.edu; mcnieves@ucdavis.edu

The L-type $Ca^{2+}$ channel $Ca_V1.2$ plays key roles in cell excitability, muscle contraction and gene expression in many cells[1,2], including arterial myocytes[3–5]. In arterial myocytes, $Ca_V1.2$ is activated by membrane depolarization and can be functionally regulated by vasoactive agents acting via protein kinase C (PKC) and protein kinase A (PKA)[6]. For example, the potent vasoconstrictor angiotensin II (ang II) activates the angiotensin type 1 receptor to stimulate L-type $Ca^{2+}$ channel activity via PKC signaling[7–10]. In contrast, extracellular elevations in glucose to levels typically found in diabetic patients and animal models of diabetes also potentiate L-type $Ca^{2+}$ channel activity via PKA-dependent phosphorylation of $Ca_V1.2$ at serine 1928 leading to vasoconstriction[11–13]. Given that PKA activation has been attributed to have a major vasorelaxant role in arterial myoyctes[14–17], a key unresolved question is how PKA signaling can also produce divergent vasoconstriction? One model is that glucose-induced PKA-mediated vasoconstriction engages specific pools of signal generators (e.g., G protein-coupled receptors), signaling enzymes (e.g., adenylyl cyclase (AC)) and effector proteins (e.g., PKA) closely associated to a specific substrate (e.g., $Ca_V1.2$)[13]. Here, activation of this signaling module could selectively enhance L-type $Ca^{2+}$ channel activity and contraction while sparing activation of other PKA-mediated pathways linked to arterial myocyte relaxation[18].

We recently discovered that glucose-induced PKA-dependent $Ca_V1.2$ phosphorylation, L-type $Ca^{2+}$ channel potentiation, and vasoconstriction required receptor-mediated subsarcolemmal cAMP production by adenylyl cyclase 5 (AC5)[19]. $P2Y_{11}$ (in humans) or $P2Y_{11}$-like (in rodents) receptors were identified as the $G_s$ protein-coupled receptor ($G_sPCR$) triggering cAMP/PKA signaling, independent of any hetero-oligomerization with $P2Y_1$ or $P2Y_6$ receptors[20,21]. Intriguingly, super-resolution microscopy revealed pools of $Ca_V1.2$ in close association with subpopulations of $P2Y_{11}/P2Y_{11}$-like, AC5 and PKA in arterial myocytes[12,19,20]. Yet, how these proteins are spatially organized to efficiently restrict glucose signaling to regulate L-type $Ca^{2+}$ channel activity and vascular reactivity is unclear.

Although initially classified as proteins that localize PKA, the multifunctional scaffold A-kinase anchoring proteins (AKAPs) are now known to orchestrate local cell signaling modules to regulate distinct cellular responses[18,22,23]. In neurons and cardiac cells, AKAP5 (referred to as AKAP150 in rodents and AKAP79 in humans) is a particularly robust signaling integrator through its interactions with AC5, PKA, PKC, calcineurin (PP2B), $Ca_V1.2$, and several G protein-coupled receptors (GPCRs)[5,24–31]. In arterial myocytes, AKAP5 has been linked to ang II- and glucose-mediated regulation of L-type $Ca^{2+}$ channels[4,11,12,32]. However, whether AKAP5 associates with purinergic receptors is unknown. Moreover, its involvement in orchestrating a $P2Y_{11}$/AC5/PKA/$Ca_V1.2$ complex to mediate regulation of L-type $Ca^{2+}$ channels upon elevated glucose in any cell type has not been examined.

Combining work in human and murine tissue/cells with a multifaceted approach that incorporates Forster resonance energy transfer (FRET) biosensors, super-resolution microscopy, proximity ligation assay, electrophysiology, ex vivo and in vivo arterial diameter, and blood flow measurements, our data provide evidence of a functional AKAP5-anchored $P2Y_{11}$/AC5/PKA/$Ca_V1.2$ nanocomplex. This complex coordinates local cAMP/PKA signaling to potentiate L-type $Ca^{2+}$ channel activity leading to vasoconstriction. We show that elements of this nanocomplex catalyze changes in cerebral blood flow upon elevations in extracellular glucose. Since elevated glucose conditions predominate in diabetes, we propose this AKAP5-anchored $P2Y_{11}$/AC5/PKA/$Ca_V1.2$ nanocomplex may be implicated in the onset of diabetic vascular complications.

## Results

### AKAP mediates cAMP signaling by glucose/NF546 in human cells.

The role of AKAP in glucose-induced, $P2Y_{11}$-dependent cAMP synthesis was examined in primary unpassaged human male and female arterial myocytes. These cells were infected with a plasma membrane (PM)-targeted Epac1-camps-based FRET biosensor (ICUE3-PM)[33,34]. The ICUE3-PM biosensor contains a Lyn sequence for PM targeting, and the FRET pair cyan fluorescent protein (CFP) and yellow fluorescent protein (YFP) attached at either side of the Epac cAMP binding site (Supplementary Fig. 1a). This sensor detects dynamic changes in cAMP production (low FRET signal) and degradation (high FRET signal)[33,34]. Targeting of the ICUE3-PM to the plasma membrane of arterial myocytes has been confirmed[19]. Human male and female arterial myocytes were obtained from adipose arteries of patients undergoing gastric sleeve surgery (Supplementary Table 1)[12,20,35]. More than 89% of these unpassaged arterial myocyte cultures are positive for smooth muscle marker and negative for endothelial, macrophages or cell lineage markers ($n = 4$ preparations; Supplementary Fig. 1b)[19,20]. These results confirm the purity of the culture system. For some experiments, cells were pretreated with the membrane-permeable ht31 peptide, which contains the PKA-anchoring domain sequence of AKAPs[36,37]. The ht31 peptide binds with high affinity to PKA, it is used to prevent AKAP-PKA interactions and thus, it disrupts AKAP function[36,37].

Pre-treatment of human arterial myocytes with 10 μM ht31 had no noticeable effect on cAMP synthesis (Fig. 1a). Application of the broad adenylyl cyclase activator forskolin elicited cAMP production in the presence of the peptide suggesting that ht31-treated cells can still synthesize cAMP. Subsequent experiments showed that elevating extracellular glucose from 5 to 15 mM induced a subtle but significant increase in cAMP in control but not in ht31-treated human arterial myocytes (Fig. 1b, d). The 5 mM and 15 mM D-glucose concentrations are similar to blood glucose levels observed in nondiabetic and diabetic patients, respectively[20,38,39], and thus justify their use as control and hyperglycemic conditions. Identical results were observed after application of the highly selective $P2Y_{11}$ agonist NF546[40] (Fig. 1c, d). In all cases, the forskolin responses were larger in magnitude (Fig. 1), suggesting that the lack of cAMP production in response to elevated glucose and NF546 in ht31-treated cells is not due to an inability of these cells to synthesize cAMP. Glucose-induced cAMP synthesis is not due to osmolarity changes as equimolar concentration of the nonpermeable mannitol or non-metabolizable L-glucose did not trigger cAMP production[19,20]. Data from human male and female arterial myocytes were comparable, and thus, were not segregated by sex. These results suggest that an AKAP mediates cAMP production upon elevated glucose and $P2Y_{11}$ receptor activation in human male and female arterial myocytes.

### Murine AKAP5 mediates cAMP signaling by glucose/NF546.

To delineate a role for AKAP5 in glucose-induced, $P2Y_{11}$-dependent cAMP production, we expressed the ICUE3-PM sensor in unpassaged aortic arterial myocytes from age-matched male wild-type (WT) and systemic AKAP5 knockout ($AKAP5^{-/-}$) mice[30,32]. The $AKAP5^{-/-}$ mice have been backcrossed into the C57BL/6J background for 10 generations[30,32]. Elevating extracellular D-glucose from 10 to 20 mM in WT arterial myocytes led to small, yet significant increases in cAMP synthesis that were magnified by forskolin (Fig. 2a, b and Supplementary Fig. 2). The D-glucose concentrations are consistent with nonfasting glucose concentrations reported in nondiabetic and diabetic mouse models, and are standard units to examine glucose-mediated changes in murine arterial myocytes[11,12,19,20,41–47]. NF546 mimicked the glucose effects (Fig. 2c, d and Supplementary Fig. 2), and no further increase

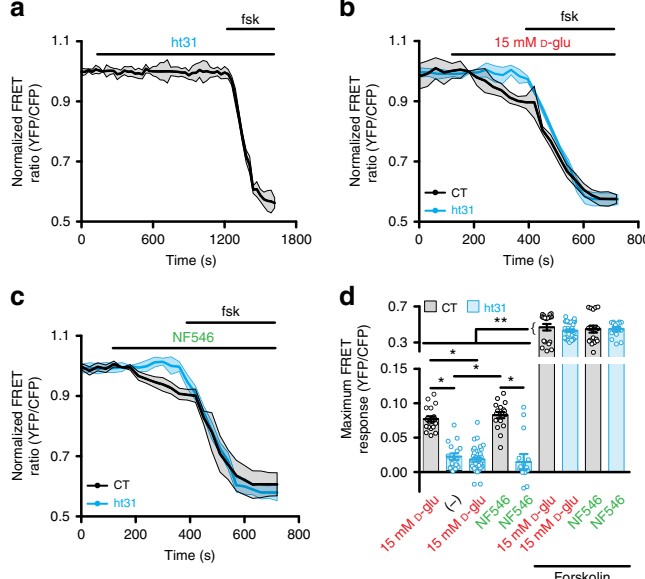

**Fig. 1 ht31 prevents glucose/NF546-induced cAMP in human cells.**
**a** Average ICUE3-PM FRET trace (mean = solid line; SEM = shade) in response to application of 10 μM ht31 and after 1 μM forskolin in human arterial myocytes. Time courses of the average ICUE3-PM FRET traces (mean = solid line; SEM = shade) in human arterial myocytes in response to **b** 15 mM D-glucose (D-glu) and **c** 500 nM NF546 and after 1 μM forskolin in control (CT) and ht31-treated cells. **d** Plot of maximum FRET responses for human male/female arterial myocytes exposed to the ht31 peptide alone ($n = 17$ cells/2 humans), 15 mM D-glu ($n = 19$ cells/3 humans for −ht31 and $n = 36$ cells/3 humans for +ht31) and NF546 ($n = 18$ cells/3 humans for −ht31 and $n = 14$ cells/3 humans for +ht31) in the absence and presence of forskolin. *$P < 0.05$ with Kruskal–Wallis one-way ANOVA with Dunn's multiple comparisons. The single asterisks highlight significant differences between all datasets in the absence of forskolin. The double asterisks indicate a statistical difference within the same experimental group in the absence and presence forskolin. $P = 0.0002$ for 15 mM D-glu-ht31 and ht31-ht31 + 15 mM D-glu + forskolin. $P = 0.0002$ for NF546-NF546 + forskolin. All other significant $P$ values are <0.0001. Data represent mean ± SEM. Source data are provided as Source data file.

occurred when 20 mM D-glucose and NF546 were added simultaneously (Fig. 2e, f and Supplementary Fig. 2). These results suggest that both compounds likely activate the same signaling pathway.

In support of the prior statement, 20 mM D-glucose + NF546-induced cAMP production in WT arterial myocytes was completely blocked by pre-treatment with the selective P2Y$_{11}$ inhibitor NF340[40] without altering the forskolin-responsive cAMP production (Supplementary Fig. 2). These results strongly implicate the involvement of a P2Y$_{11}$-like receptor in this process. In agreement with a role for AKAP5, application of 20 mM D-glucose, NF546, or 20 mM D-glucose + NF546 failed to induce cAMP synthesis in AKAP5$^{−/−}$ cells (Fig. 2). This was not due to changes in protein expression of P2Y$_{11}$-like receptor, AC5, PKA, or Ca$_V$1.2[47] in AKAP5$^{−/−}$ arterial lysates compared to WT (Supplementary Fig. 3) or an inability of AKAP5$^{−/−}$ arterial myocytes to produce cAMP as forskolin response remained intact in these cells (Fig. 2). These results suggest a role for AKAP5 in spatially confining cAMP synthesis in response to elevated glucose and activation of the P2Y$_{11}$-like receptor in mouse arterial myocytes.

**Glucose/NF546 boost I$_{Ba}$ and vasoconstriction via AKAP5/AC5.**
Having determined that AKAP5 organizes both glucose and

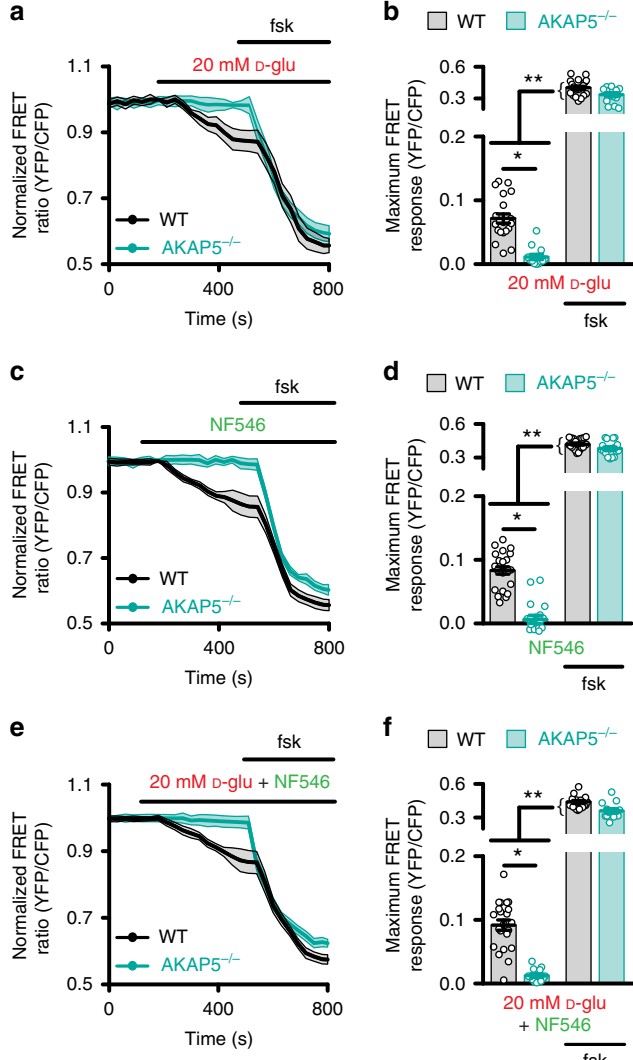

**Fig. 2 AKAP5 is necessary for glucose/NF546-induced cAMP synthesis.**
Averaged ICUE3-PM FRET traces (mean = solid line; SEM = shade) and plot of maximum FRET ratio for wild-type (WT) and AKAP5$^{−/−}$ male arterial myocytes in response to **a**, **b** 20 mM D-glucose (D-glu; $n = 21$ cells/6 WT mice; $n = 13$ cells/6 AKAP5$^{−/−}$ mice), **c**, **d** 500 nM NF546 ($n = 23$ cells/6 WT mice; $n = 18$ cells/6 AKAP5$^{−/−}$ mice), and **e**, **f** 20 mM D-glu + NF546 ($n = 22$ cells/6 WT mice; $n = 13$ cells/6 AKAP5$^{−/−}$ mice) and 1 μM forskolin. *$P < 0.05$ with two-tailed Mann–Whitney test. The single asterisks highlight significant differences between datasets in the absence of forskolin. The double asterisks indicate a statistical difference within the same experimental group in the absence and presence forskolin. All significant $P$ values are <0.0001. Data represent mean ± SEM. Source data are provided as Source data file.

NF546-induced cAMP signaling, we next sought to examine whether AKAP5 couples the cAMP synthesis to potentiation of L-type Ca$^{2+}$ channels and vasoconstriction in response to these stimuli. First, we used whole-cell patch-clamp electrophysiology with barium (Ba$^{2+}$) as the charge carrier to record L-type Ca$^{2+}$ channel-mediated currents (I$_{Ba}$). We established that 20 mM D-glucose potentiated I$_{Ba}$ compared to 10 mM D-glucose in freshly dissociated WT male cerebral arterial myocytes (Supplementary Fig. 4a, b). Yet, this response was absent in AKAP5$^{−/−}$ male cells. Similar results were observed in WT and AKAP5$^{−/−}$ female cerebral arterial myocytes (Supplementary Fig. 4c, d). These results suggest that elevated glucose potentiates L-type Ca$^{2+}$

channel activity via an AKAP5-dependent mechanism in male and female arterial myocytes.

To assess the physiological relevance of the AKAP5-dependent regulation of L-type $Ca^{2+}$ channels upon elevation in extracellular glucose, we measured myogenic tone in pressurized (60 mmHg) middle cerebral arteries isolated from male WT and $AKAP5^{-/-}$ mice. Only arteries that robustly constricted to a depolarizing stimulus (i.e., 60 mM $K^+$) and developed spontaneous tone upon increasing intravascular pressure from 20 to 60 mmHg were considered for analysis. Elevating glucose from 10 to 20 mM caused robust constriction of WT cerebral arteries (Figs. 3a, b and Supplementary Table 2). Similar results are observed in endothelium-denuded arteries[12]. In contrast, 20 mM D-glucose failed to constrict $AKAP5^{-/-}$ arteries (Fig. 3a, b and Supplementary Table 2). WT and $AKAP5^{-/-}$ arteries had similar

responses to 60 mM $K^+$, suggesting that vasoconstriction was not impaired due to AKAP5 depletion ($P = 0.620$ with two-tailed Mann–Whitney test; Supplementary Table 2). The change in myogenic tone in response to 20 mM D-glucose was significantly larger in WT arteries compared to $AKAP5^{-/-}$ vessels (Fig. 3c and Supplementary Table 2). These data indicate that AKAP5 is necessary for enhanced L-type $Ca^{2+}$ channel activity and vasoconstriction upon elevated glucose.

We investigated whether AKAP5 and AC5 are required for regulation of L-type $Ca^{2+}$ channels and vascular reactivity upon direct activation of the murine $P2Y_{11}$-like receptor. We examined AC5 because it has been implicated in this pathway[19]. To test the role of AC5, we used arterial myocytes and arteries from age-matched AC5 knockout ($AC5^{-/-}$) mice that have been backcrossed for 10 generations[19,48]. Initial experiments using unpassaged

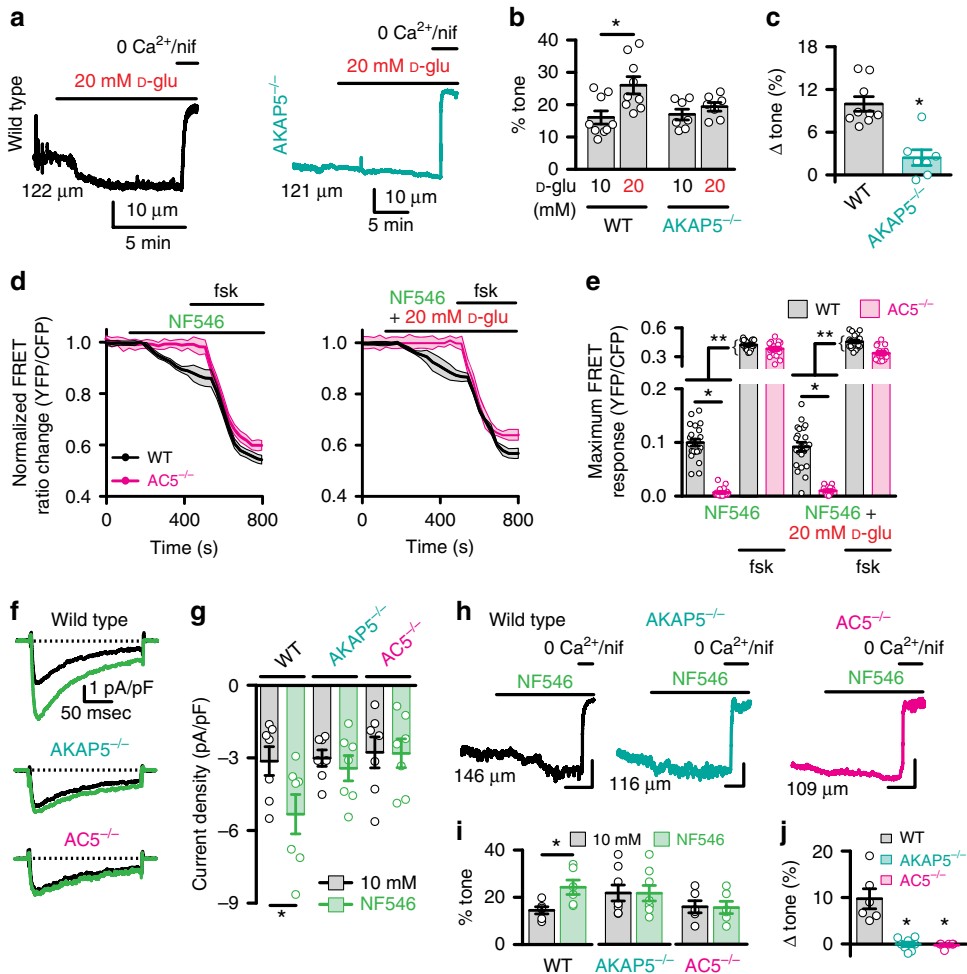

**Fig. 3 AKAP5/AC5 mediate $I_{Ba}$ potentiation and vasoconstriction upon glucose/NF546.** Representative **a** diameter recordings, **b** summary percentage myogenic tone, and **c** change in myogenic tone from wild-type (WT; $n = 9$ arteries/5 mice) and $AKAP5^{-/-}$ ($n = 7$ arteries/6 mice) cerebral arteries in 10 and 20 mM D-glucose (D-glu). *$P < 0.05$ with two-tailed Wilcoxon test for **b** ($P = 0.0039$ for WT 10 mM D-glu-20 mM D-glu) and two-tailed Mann–Whitney test for **c** ($P = 0.0012$ for WT-$AKAP5^{-/-}$). **d** Averaged ICUE3-PM traces (mean = solid line; SEM = shade) and **e** maximum FRET ratio for WT and $AC5^{-/-}$ arterial myocytes upon 500 nM NF546 ($n = 23$ cells/6 WT mice; $n = 18$ cells/5 $AC5^{-/-}$ mice), 20 mM D-glu + 500 nM NF546 ($n = 22$ cells/6 WT mice; $n = 15$ cells/5 $AC5^{-/-}$ mice) and after 1 µM forskolin. *$P < 0.05$ with two-tailed Mann–Whitney test. Single asterisks highlight significant differences between datasets in the absence of forskolin. Double asterisks indicate statistical differences within the same experimental group before and after forskolin. All significant $P$ values are <0.0001. **f** Exemplary $I_{Ba}$ traces and **g** amalgamated current density from WT ($n = 7$ cells/5 mice), $AKAP5^{-/-}$ ($n = 7$ cells/5 mice) and $AC5^{-/-}$ ($n = 7$ cells/3 mice) cerebral arterial myocytes before and after 500 nM NF546. *$P < 0.05$ with two-tailed Wilcoxon test. $P = 0.0156$ for WT 10 mM D-glu-NF546. **h** Representative diameter recordings, **i** summary percentage myogenic tone, and **j** change in myogenic tone from WT ($n = 6$ arteries/4 mice), $AKAP5^{-/-}$ ($n = 8$ arteries/6 mice), and $AC5^{-/-}$ ($n = 6$ arteries/4 mice) cerebral arteries exposed to 500 nM NF546. Scales = 10 µm (vertical) and 5 min (horizontal). *$P < 0.05$ with two-tailed Wilcoxon test for Dii ($P = 0.0313$ for WT 10 mM D-glu-NF546) and Kruskal–Wallis one-way ANOVA with Dunn's multiple for Diii ($P = 0.0036$ for WT-$AKAP5^{-/-}$ and $P = 0.0040$ for WT-$AC5^{-/-}$). Initial diameters shown on the traces left side. Data represent mean ± SEM. Source data are provided as Source data file.

cultured WT and AC5$^{-/-}$ arterial myocytes expressing the ICUE3-PM sensor revealed that AC5 depletion prevented cAMP synthesis upon application of the P2Y$_{11}$ agonist NF546 or NF546 + 20 mM D-glucose compared to WT cells (Fig. 3d, e). Yet, forskolin induced a larger cAMP synthesis in all experimental conditions that was of similar magnitude in WT and AC5$^{-/-}$ cells. These results suggest that AC5 is essential for cAMP production in response to elevated glucose or NF546 in mouse arterial myocytes.

Next, we established that application of the P2Y$_{11}$ agonist NF546 stimulated I$_{Ba}$ in WT but not AKAP5$^{-/-}$ and AC5$^{-/-}$ arterial myocytes (Fig. 3f, g). Subsequently, pressure myography showed that NF546-induced vasoconstriction in WT arteries was hindered in AKAP5$^{-/-}$ and AC5$^{-/-}$ arteries (Fig. 3h, i and Supplementary Table 2). The 60 mM K$^+$ responses were similar in WT, AKAP5$^{-/-}$, and AC5$^{-/-}$ arteries, thus ratifying viability of the arteries ($P = 0.3823$ with Kruskal–Wallis one-way ANOVA with Dunn's multiple comparisons; Supplementary Table 2). The change in myogenic tone in response to NF546 was significantly larger in WT compared to AKAP5$^{-/-}$ and AC5$^{-/-}$ arteries (Fig. 3j and Supplementary Table 2). These results suggest a critical role for AKAP5 and AC5 in potentiation of L-type Ca$^{2+}$ channels and vasoconstriction by glucose and NF546 stimulation.

**AKAP5 regulates myogenic tone and blood flow in vivo**. To evaluate the role of AKAP5 on myogenic tone and blood flow in response to elevated glucose and NF546 in an in vivo setting, we fitted anesthetized mice with a cranial window[19]. With this procedure, arterial diameter and relative blood flow dynamics can be recorded using intravital microscopy and laser speckle imaging, respectively[49,50]. Data were acquired before and after the topical treatment of middle cerebral arteries and branches.

Arterial diameter was stable when the cranial window was treated with 10 mM D-glucose. Permeation of the cranial window with 20 mM D-glucose, but not 20 mM mannitol, constricted WT cerebral arteries resulting in an increase in myogenic tone (Fig. 4a, b, Supplementary Fig. 5a and Supplementary Table 3). Simultaneous application of 20 mM D-glucose + NF546 did not change the magnitude of myogenic tone compared to 20 mM D-glucose alone (Fig. 4a, b, Supplementary Fig. 5a and Supplementary Table 3). These results suggest that elevated glucose and NF546 likely act through the same mechanism in vivo. Subsequent analysis of relative blood flow maps obtained with laser speckle imaging revealed a decrease in blood flow in WT arteries in response to 20 mM D-glucose and 20 mM D-glucose + NF546, compared to 10 mM mannitol (Fig. 4c, d). In contrast, ablation of AKAP5 prevented changes in myogenic tone and blood flow in response to 20 mM D-glucose and 20 mM D-glucose + NF546 when compared to the 10 mM mannitol condition (Fig. 4a–d, Supplementary Fig. 5a and Supplementary Table 3).

To further confirm a role for P2Y$_{11}$-like receptor in glucose and NF546 modulation of myogenic tone in vivo, we performed a series of experiments in which the cranial window in WT mice was pretreated with the P2Y$_{11}$ inhibitor NF340 (10 μM). Under control conditions (10 mM D-glucose), NF340 had no effect on basal arterial diameter and therefore myogenic tone (Fig. 4e, f, Supplementary Fig. 5b and Supplementary Table 3). Meanwhile, NF340 blocked vasoconstriction and any change in myogenic tone in response to 20 mM D-glucose or 20 mM D-glucose + NF546 (Fig. 4e, f, Supplementary Fig. 5b and Supplementary Table 3). Collectively, these results suggest that in vivo changes in myogenic tone and blood flow in response to elevated glucose require AKAP5 and are mediated by activation of the P2Y$_{11}$-like receptor in mouse cerebral arteries.

**Nanometer proximity between AKAP5, P2Y$_{11}$, AC5, and Ca$_V$1.2**. We have shown close clustering/association between subpopulations of Ca$_V$1.2 and PKA[12,20], Ca$_V$1.2 and AC5[19], and Ca$_V$1.2 and P2Y$_{11}$[20] in arterial myocytes. To investigate the spatial relationship between AKAP5, P2Y$_{11}$-like receptors, AC5, and Ca$_V$1.2 in these cells, we first fixed and performed triple labeling of arterial myocytes with the following combination of antibodies: (1) AKAP5/P2Y$_{11}$/Ca$_V$1.2 or (2) AKAP5/AC5/Ca$_V$1.2. Antibodies for these proteins have been validated by our group and commercial sources (refs. [12,19,20,32,35,47]; see also Supplementary Table 5). Images of triple-labeled arterial myocytes were collected using a super-resolution confocal microscope coupled with an Airyscanner module. This system has an axial resolution of ~120 nm. Intensity projection images of arterial myocytes and line profile analysis from these images showed adjacent and/or overlapping fluorescence associated with AKAP5, P2Y$_{11}$, and Ca$_V$1.2 (Fig. 5a, b). We noted that the line intensity profile for P2Y$_{11}$ in Fig. 5ii had a less tight matching contour compared to line profiles for AKAP5 and Ca$_V$1.2. This is likely because the P2Y$_{11}$-associated fluorescence obtained with the Airyscan shows a more fragmented plasma membrane distribution. Yet, data show a number of areas where there is a clear coincidence of the three colors, thus suggesting close proximity between a subset of these three proteins. A similar fluorescence pattern was observed in cells stained for AKAP5, AC5, and Ca$_V$1.2 (Fig. 5c, d). No fluorescence signals were detected in cells incubated with rabbit/mouse/goat nonimmune IgGs (Supplementary Fig. 6a).

To further explore and quantify the close association between AKAP5, P2Y$_{11}$-like receptors, AC5, and Ca$_V$1.2 in mouse arterial myocytes at a higher spatial resolution, we used ground state depletion (GSD) super-resolution nanoscopy in the total internal reflection fluorescence (TIRF) mode. This technique allows the recording of protein clusters at/near the plasma membrane with a higher lateral resolution of about 20–40 nm[12,51]. Fluorescence signals were not detected from cells in which the primary antibodies were omitted (Supplementary Fig. 6b) or in which nonimmune IgGs were used[19]. GSD reconstruction maps for AKAP5, P2Y$_{11}$, AC5, and Ca$_V$1.2 in arterial myocytes can be appreciated in the upper panels of Fig. 5e–g. Magnified images of two areas within the cells revealed that these proteins form clusters at the plasma membrane of arterial myocytes of various sizes and densities (lower panels in Figs. 5e–g and Supplementary Fig. 6c–h). Line profile analysis and merged maps of AKAP5/P2Y$_{11}$, AKAP5/AC5, and AKAP5/Ca$_V$1.2 suggest a close association between subsets of these proteins (Fig. 5h–j). Distance histograms of AKAP5 to nearest P2Y$_{11}$, AC5, or Ca$_V$1.2 obtained using a nearest-neighbor analysis revealed two components with the closest centroid of AKAP5-P2Y$_{11}$, AKAP5-AC5, and AKAP5-Ca$_V$1.2 at 40, 44, and 42 nm, respectively (Fig. 5k–m). To examine if the close association between AKAP5, P2Y$_{11}$, AC5, and Ca$_V$1.2 is mediated by a specific organization between these proteins, we compared the percentage of overlap between AKAP5-P2Y$_{11}$, AKAP5-AC5, and AKAP5-Ca$_V$1.2 using the experimental super-resolution localization maps and randomized images. The randomized images of AKAP5, P2Y$_{11}$, AC5, and Ca$_V$1.2 were obtained from data derived from the experimental super-resolution localization maps for these protein pairs using the Coste's randomization algorithm in the ImageJ JACoP plug-in[20,52]. Our analysis showed that the percentage of overlap between AKAP5-P2Y$_{11}$, AKAP5-AC5, and AKAP5-Ca$_V$1.2 obtained from the experimental super-resolution localization maps was significantly higher than that observed for the simulated random distribution between these proteins (Supplementary Fig. 6i–n). Altogether, these results suggest close spatial proximity between AKAP5, P2Y$_{11}$-like receptors, AC5, and Ca$_V$1.2.

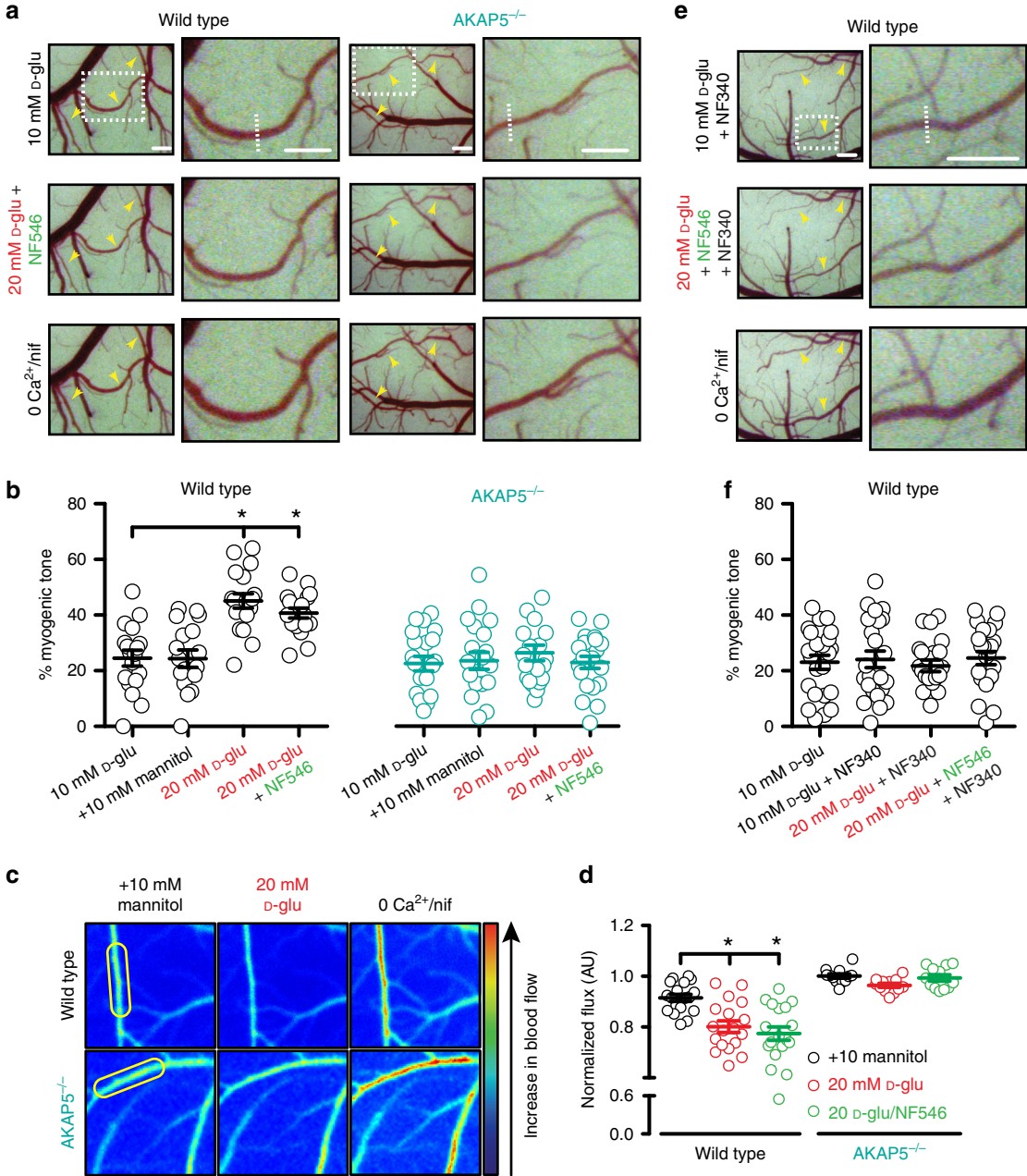

**Fig. 4 Glucose/NF546 effects on myogenic tone/blood flow require AKAP5. a** Exemplary open cranial window images of wild-type (WT; $n = 18$ arteries/3 mice) and AKAP5$^{-/-}$ ($n = 21$ arteries/3 mice) middle cerebral arteries and branches exposed to 10 mM D-glucose (D-glu), 20 mM D-glu + NF546, and 0 Ca$^{2+}$ + 1 μM nifedipine (nif). **b** Summary myogenic tone data. *$P < 0.05$ with Friedman one-way ANOVA with Dunn's multiple comparisons. Significance was compared to the 10 mM D-glu group. $P < 0.0001$ for WT 10 mM D-glu-20 mM D-glu and $P = 0.0004$ for WT 10 mM D-glu-20 mM D-glu + NF546. **c** Representative pseudo•colored blood flow images of cerebral pial arteries through a cranial window exposed to 10 mM D-glu + 10 mM mannitol, 20 mM D-glu, and 0 Ca$^{2+}$ + 1 μM nif in WT and AKAP5$^{-/-}$ mice. **d** Summary blood flow data in WT ($n = 18$ arteries/5 mice) and AKAP5$^{-/-}$ ($n = 12$ arteries/3 mice) mice. *$P < 0.05$ with Friedman one-way ANOVA with Dunn's multiple comparisons. Significance was compared to the +10 mM mannitol group. $P = 0.0007$ for WT mannitol-20 mM D-glu and $P < 0.0001$ for WT mannitol-20 mM D-glu + NF546. The yellow ovals highlight regions that were used for analysis. **e** Representative open cranial window images of WT ($n = 23$ arteries/3 mice) middle cerebral arteries and branches in 10 mM D-glu + 10 μM NF340, 20 mM D-glu + 500 nM NF546 + 10 μM NF340, and 0 Ca$^{2+}$ + 1 μM nif. **f** Amalgamated myogenic tone data. *$P < 0.05$, Friedman one-way ANOVA with Dunn's multiple comparisons. Significance was compared to the 10 mM D-glucose group. Data represent mean ± SEM. In (**a**) and (**e**), yellow arrows = arteries, dotted rectangles highlight zoom regions on the right side panels, and dotted lines in the zoom images highlight regions that were used for analysis. Scale bars = 50 pixels for the main figures and 70 pixels for the insets. Source data are provided as Source data file.

**AKAP5 orchestrates the P2Y$_{11}$/AC5/PKA/Ca$_V$1.2 nanocomplex.**
To complement and expand on the super-resolution data, we used the proximity ligation assay (PLA)[53]. This assay emits a fluorescent signal only when proteins of interest are at least within 40 nm of each other[53] (as extensively validated by these authors[12,19,20,35]). In

freshly dissociated human male and female arterial myocytes, PLA data show close association between AKAP5-Ca$_V$1.2 and PKA$_{cat}$-Ca$_V$1.2 (Supplementary Fig. 7a–c). We also observed a robust PLA signal between AKAP5-P2Y$_{11}$, AKAP5-AC5, and P2Y$_{11}$-AC5 (Supplementary Fig. 7a–c), which suggests the formation of a

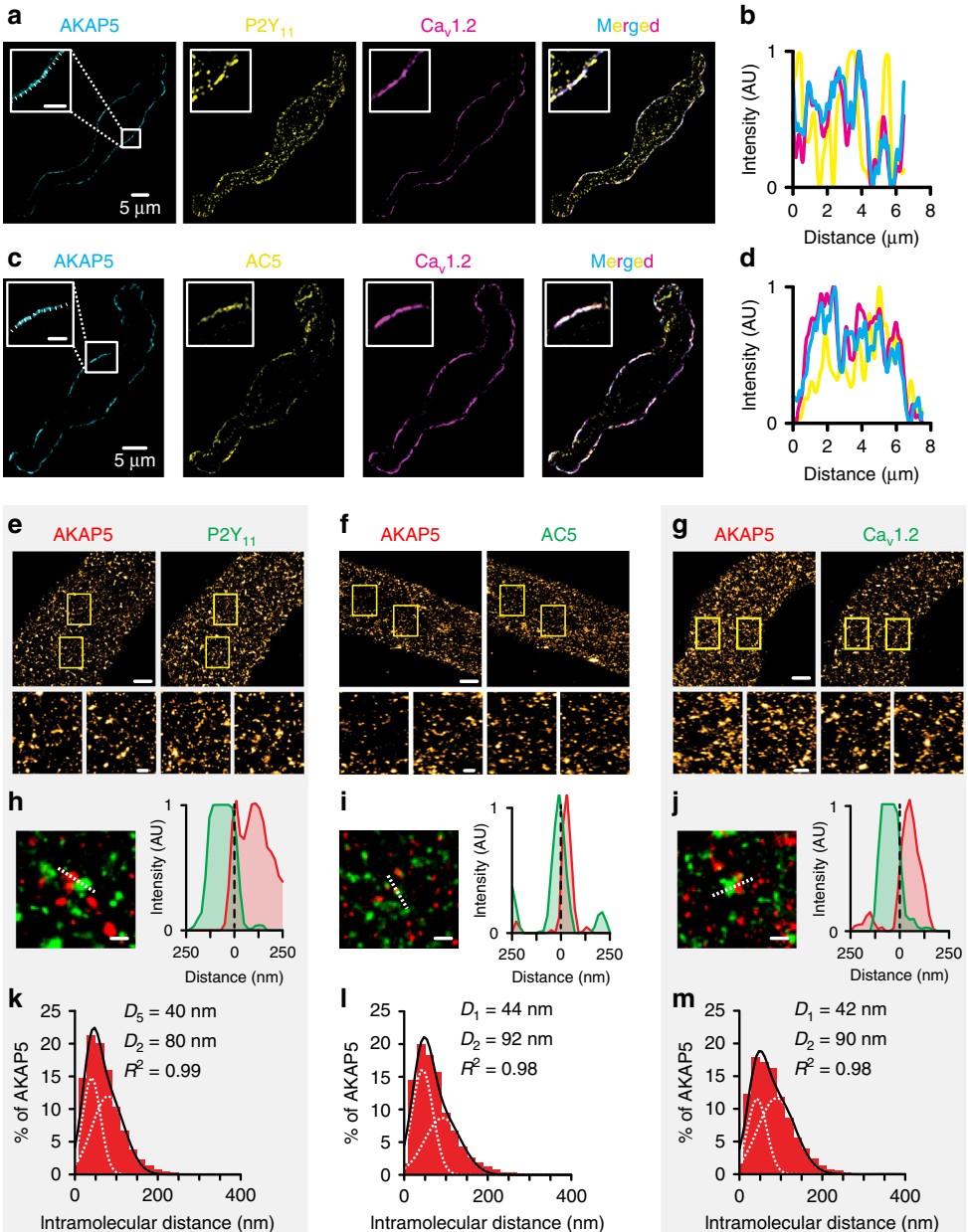

**Fig. 5 AKAP5 closely associates with P2Y$_{11}$-like, AC5, and Ca$_V$1.2.** Representative super-resolution Airyscan intensity projection images and line intensity profile of arterial myocytes triple-labeled for either **a**, **b** AKAP5 (cyan), P2Y$_{11}$ (yellow), and Ca$_V$1.2 (magenta) or **c**, **d** AKAP5 (cyan), AC5 (yellow), and Ca$_V$1.2 (magenta). Merged images are on the right side panels. The magnified insets are from regions highlighted by the white squares in the AKAP5 images. Insets' scale bars = 2 μm. Line intensity profiles obtained from the regions below the dotted lines in the magnified insets in (**a**) and (**c**). Similar results were observed in six different cells from two independent preparations per condition. Representative super-resolution GSD reconstruction maps of arterial myocytes labeled for **e** AKAP5 and P2Y$_{11}$, **f** AKAP5 and AC5, or **g** AKAP5 and Ca$_V$1.2 (scale bar = 2 μm). Two magnified areas obtained from the region highlighted by the yellow boxes in the main figure are shown (scale bar = 200 nm). These magnified panels are intended to provide a clearer view of cluster size and distribution in two areas of a cell. Enlarged merged images (left) and associated x–y fluorescence intensity profile (right) from the areas highlighted by the dotted lines of areas of close proximity between **h** AKAP5 and P2Y$_{11}$, **i** AKAP5 and AC5, or **j** AKAP5 and Ca$_V$1.2 (scale bar = 200 nm). Histograms of the lowest intermolecular distance to AKAP5 centroids for **k** P2Y$_{11}$ ($n = 11,778$ particles from 4 cells/2 WT mice), **l** AC5 ($n = 10,049$ particles from 5 cells/2 WT mice), or **m** Ca$_V$1.2 ($n = 14,750$ particles from 5 cells/2 WT mice) fluorescence particles. Data were fit with a sum of two Gaussian functions with depicted centroids and R$^2$. Source data are provided as Source data file.

complex containing AKAP5, a purinergic receptor and AC5 in excitable cells. PLA signal was not observed when only one primary antibody was present (Supplementary Fig. 7b, c). Together with prior studies showing a close association between PKA/Ca$_V$1.2, P2Y$_{11}$/Ca$_V$1.2, and P2Y$_{11}$/PKA in human male and female arterial myocytes[12,20], these results suggest that AKAP5, P2Y$_{11}$, AC5, PKA,

and Ca$_V$1.2 form a nanocomplex in human arterial myocytes that is consistent between sexes.

A similar association pattern was observed in WT mouse arterial myocytes. PLA signals were detected in cells co-labeled for P2Y$_{11}$-Ca$_V$1.2, P2Y$_{11}$-PKA$_{cat}$, P2Y$_{11}$-AKAP5, Ca$_V$1.2-AKAP5, AC5-AKAP5, AC5-P2Y$_{11}$, AC5-PKA$_{cat}$, and Ca$_V$1.2-PKA$_{RIIα}$

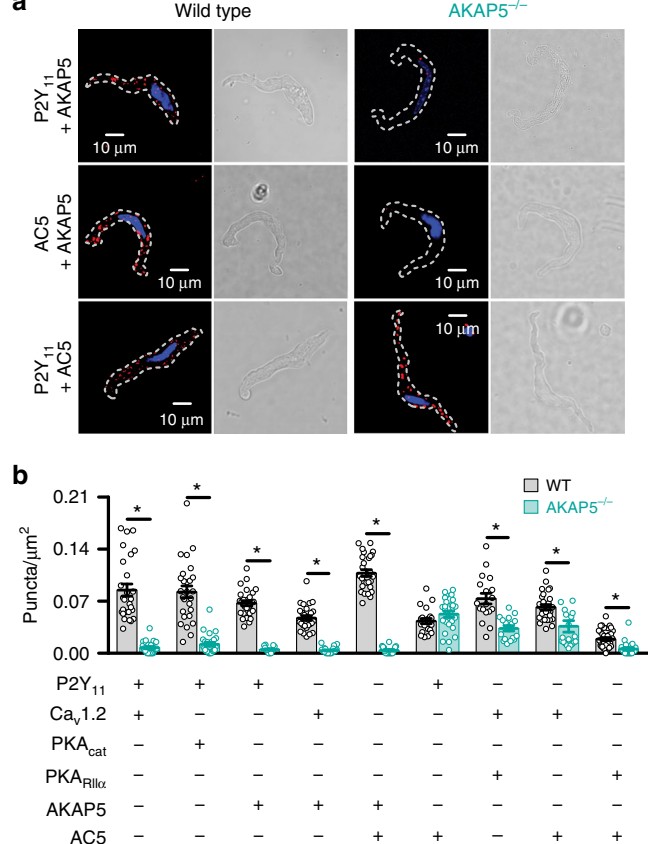

**Fig. 6 AKAP5 fosters the P2Y$_{11}$, AC5, PKA, and Ca$_V$1.2 nanocomplex.**
**a** Exemplary merged fluorescence proximity ligation assay (PLA; red)/nucleus (blue; left panels) and differential interference contrast (right panels) images of wild-type (WT) and AKAP5$^{-/-}$ cerebral arterial myocytes co-labeled for P2Y$_{11}$ + AKAP5, AC5 + AKAP5, and P2Y$_{11}$ + AC5. **b** Quantification of proximity ligation assay (PLA) fluorescent puncta per cell area (puncta/μm$^2$) for WT and AKAP5$^{-/-}$ cerebral arterial myocytes labeled for P2Y$_{11}$ + Ca$_V$1.2 ($n = 27$ cells/3 WT mice; $n = 26$ cells/3 AKAP5$^{-/-}$ mice), P2Y$_{11}$ + PKA$_{cat}$ ($n = 27$ cells/3 WT mice; $n = 26$ cells/3 AKAP5$^{-/-}$ mice), P2Y$_{11}$ + AKAP5 ($n = 28$ cells/3 WT mice; $n = 15$ cells/3 AKAP5$^{-/-}$ mice), Ca$_V$1.2 + AKAP5 ($n = 27$ cells/3 WT mice; $n = 24$ cells/3 AKAP5$^{-/-}$ mice), AC5 + AKAP5 ($n = 29$ cells/3 WT mice; $n = 18$ cells/3 AKAP5$^{-/-}$ mice), P2Y$_{11}$ + AC5 ($n = 22$ cells/3 WT mice; $n = 27$ cells/3 AKAP5$^{-/-}$ mice), Ca$_V$1.2 + PKA$_{RII\alpha}$ ($n = 18$ cells/3 WT mice; $n = 17$ cells/3 AKAP5$^{-/-}$ mice), Ca$_V$1.2 + AC5 ($n = 28$ cells/3 WT mice; $n = 20$ cells/3 AKAP5$^{-/-}$ mice), and PKA$_{RII\alpha}$ + AC5 ($n = 36$ cells/3 WT mice; $n = 22$ cells/3 AKAP5$^{-/-}$ mice). *$P < 0.05$ with two-tailed Mann–Whitney test. Comparison between WT and AKAP5$^{-/-}$ pairs. $P = 0.0003$ for Ca$_V$1.2-AC5, and all other significant $P$ values are <0.0001. Data represent mean ± SEM. Source data are provided as Source data file.

(Fig. 6a, b and Supplementary Fig. 8). In contrast to WT cells, the prominent associations between P2Y$_{11}$-Ca$_V$1.2, P2Y$_{11}$-PKA$_{cat}$, AC5-PKA$_{cat}$, and Ca$_V$1.2-PKA$_{RII\alpha}$ were abolished or significantly reduced in AKAP5$^{-/-}$ arterial myocytes (Fig. 6a, b and Supplementary Fig. 8). Negative controls in which one primary antibody from each combination was omitted or when non-immune IgGs were used as primary antibodies displayed no puncta in both WT and AKAP5$^{-/-}$ arterial myocytes (Supplementary Fig. 9a, b). Intriguingly, the density of PLA puncta for cells co-labeled for P2Y$_{11}$ and AC5 was similar in WT and AKAP5$^{-/-}$ arterial myocytes, suggesting that AKAP5 may not be necessary for a close association of these two proteins (Fig. 6b). Altogether, these results suggest that AKAP5 plays a crucial role

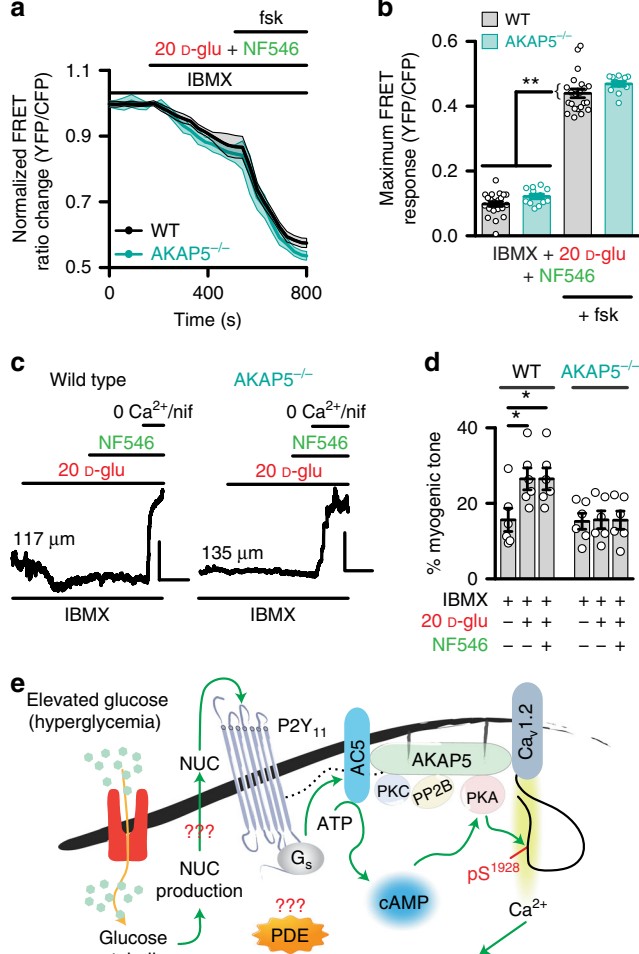

**Fig. 7 PDE inhibition restores cAMP signal but not vasoconstriction in AKAP5$^{-/-}$ tissue. a** Time courses of the average ICUE3-PM traces (mean = solid line; SEM = shade) and **b** summary data of wild-type (WT; $n = 22$ cells/3 mice) and AKAP5$^{-/-}$ ($n = 12$ cells/3 mice) arterial myocytes pretreated with the broad phosphodiesterase (PDE) inhibitor 3-isobutyl-1-methylxanthine (IBMX; 100 μM) in response to 20 mM D-glucose (D-glu) + NF546 and after application of forskolin. *$P < 0.05$ with two-tailed Mann–Whitney test. The single asterisks highlight significant differences between datasets in the absence of forskolin. The double asterisks indicate a statistical difference within the same experimental group in the absence and presence forskolin. All significant $P$ values are <0.0001. **c** Exemplary diameter recordings and **d** summary myogenic tone data from WT ($n = 6$ arteries/4 mice) and AKAP5$^{-/-}$ ($n = 6$ arteries/3 mice) cerebral arteries pretreated with IBMX (100 μM) before and after application of 20 mM D-glu and subsequent NF546. Scales = 10 μm (vertical) and 5 min (horizontal). Initial diameters shown on the traces left side. *$P < 0.05$ with Friedman one-way ANOVA with Dunn's multiple comparisons. Statistical differences were compared between datasets within the same phenotype. $P = 0.0281$ for WT 10 mM D-glu + IBMX-20 mM D-glu + IBMX, and $P = 0.0281$ for WT 10 mM D-glu + IBMX-20 mM D-glu + NF546 + IBMX. Data represent mean ± SEM. **e** Proposed model for an AKAP5-dependent signaling module formed by P2Y$_{11}$/P2Y$_{11}$-like receptor, AC5, PKA and Ca$_V$1.2, which regulates L-type Ca$^{2+}$ channel activity via Ca$_V$1.2 serine 1928 phosphorylation (pS$^{1928}$). Glucose-induced potentiation of L-type Ca$^{2+}$ channel activity promotes vasoconstriction. The dotted black line is to reflect close association between AKAP5 and P2Y$_{11}$. PP2B = calcineurin. ATP = adenosine triphosphatase. The mechanisms of PDE regulation and nucleotide (NUC) release remain unclear as denoted by the red question mark symbols. Source data are provided as Source data file.

in spatially organizing an AKAP5/P2Y$_{11}$/P2Y$_{11}$-like receptor/AC5/PKA/Ca$_V$1.2 nanodomain.

**PDEs regulates cAMP and vasoconstriction in AKAP5$^{-/-}$ tissue.** The results above raised the question of why glucose/NF546-induced cAMP synthesis is completely eliminated in cells treated with ht31 or from AKAP5$^{-/-}$ mice if the P2Y$_{11}$/P2Y$_{11}$-like receptor and AC5 remains in close proximity of each other? Prior studies have suggested that disruption of AKAP function may alter the activity of signaling proteins, including phosphodiesterases (PDEs)[25,54,55]. Consistent with this possibility, we found that WT and AKAP5$^{-/-}$ cells pretreated with the broad PDE inhibitor 3-isobutyl-1-methylxanthine (IBMX; 100 μM) showed localized cAMP production of the same magnitude in response to 20 mM D-glucose, NF546, and 20 mM D-glucose + NF546 (Fig. 7a, b and Supplementary Fig. 10). These results suggest that genetic ablation of AKAP5 alters PDE activity in arterial myocytes to prevent glucose/NF546-induced cAMP synthesis. Intriguingly, whereas pressurized WT arteries pretreated with IBMX (100 μM) constricted to the application of 20 mM D-glucose and NF546, this response was completely lost in IBMX-pretreated AKAP5$^{-/-}$ arteries, even when glucose/NF546-induced cAMP signaling is restored (Fig. 7c, d and Supplementary Table 4). The 60 mM K$^+$ response was similar in IBMX-treated WT and AKAP5$^{-/-}$ arteries ($P = 0.2403$ with two-tailed Mann–Whitney test; Supplementary Table 4), thus indicating that differences between WT and AKAP5$^{-/-}$ arteries are not due to an inability of AKAP5$^{-/-}$ vessels to constrict. Together, results suggest that an intact AKAP5/P2Y$_{11}$-like receptor/AC5/PKA/Ca$_V$1.2 nanocomplex is necessary for regulation of vascular reactivity in response to elevated glucose and the P2Y$_{11}$ agonist NF546.

## Discussion

In this study, we report three key findings regarding an unforeseen role for AKAP5 in functionally orchestrating a purinergic signaling nanocomplex (Fig. 7e). First, we found that AKAP5 is necessary for the assembly of a signaling unit containing P2Y$_{11}$/P2Y$_{11}$-like receptors, AC5, PKA, and Ca$_V$1.2 in human and mouse arterial myocytes. Second, an AKAP in human and AKAP5 in mouse arterial myocytes is critical for cAMP synthesis in response to elevated glucose and direct activation of the P2Y$_{11}$/P2Y$_{11}$-like receptors. Third, the AKAP5-driven nanocomplex is essential for stimulation of L-type Ca$^{2+}$ channel activity, enhanced vasoconstriction and changes in blood flow upon elevated glucose and NF546. Altogether, these findings highlight an AKAP5-driven signaling nanocomplex that has profound influence in the control of vascular function in response to elevated extracellular glucose. Given that hyperglycemia is a major metabolic abnormality in diabetes, we propose that this AKAP5/P2Y$_{11}$/AC5/PKA/Ca$_V$1.2 nanocomplex could be involved in the development of diabetic vascular complications.

Coupling of distinct AKAP-mediated nanocomplexes to specific AC isoforms and binding partners offer a molecular blueprint for spatial segregation and organization of cAMP signaling[18,22]. Such distinct and compartmentalized signaling nanocomplexes enable local cAMP production to encode disparate responses within the same tissue (e.g., contraction versus relaxation by PKA activation in arterial myocytes). Supporting this idea, we observed in human and mouse arterial myocytes subtle, yet significant increases in cAMP synthesis in response to elevated glucose, indicating localized cAMP production (Figs. 1 and 2, and Supplementary Fig. 2). This observation contrasted with the global cAMP synthesis induced by forskolin. Consistent with a role for AKAP5 and AC5 signaling in this process, we

show that glucose-, but not forskolin-induced cAMP synthesis was hampered in AKAP5$^{-/-}$ and AC5$^{-/-}$ arterial myocytes. These observations tally with results obtained in human cells exposed to the ht31 peptide. Data also implicate AKAP5 and AC5 as being essential for localized cAMP synthesis after activation of P2Y$_{11}$/P2Y$_{11}$-like receptors[20]. We thus propose that AKAP5 and AC5 facilitate the spatial confinement of cAMP signaling triggered by activation of P2Y$_{11}$/P2Y$_{11}$-like receptors upon elevated glucose in arterial myocytes.

Here, evidence is provided for the formation of an AKAP5-driven nanocomplex that organizes P2Y$_{11}$/P2Y$_{11}$-like receptors, AC5, PKA, and Ca$_V$1.2 in arterial myocytes (Fig. 7e). This view was consolidated by using powerful approaches such as super-resolution imaging and proximity ligation assay. Accordingly, we conclude that a close association between AKAP5 with P2Y$_{11}$/P2Y$_{11}$-like receptors and AC5, as well as AC5 with P2Y$_{11}$/P2Y$_{11}$-like receptors and PKA occurs in human and mouse arterial myocytes (Figs. 5 and 6 and Supplementary Figs. 6 and 7). These findings highlight previously unappreciated molecular interactions between purinergic receptors (i.e., P2Y$_{11}$) and AKAP5. Moreover, the close proximity between P2Y$_{11}$ and AC5 functionally couples the upstream G$_s$-coupled purinergic receptor to an essential AC isoform that drives downstream glucose effects on vascular L-type Ca$^{2+}$ channels and vascular reactivity[19,20]. This nanocomplex may have broad (patho)physiological implications in conditions such as diabetic vasculopathies. Indeed, recent data indicate that enhanced close association between key members of this nanocomplex (i.e., AC5, PKA, and Ca$_V$1.2) mediates enhanced vascular L-type Ca$^{2+}$ channel activity and myogenic tone during diabetes[12,19]. These data suggest that the complex could remain together, and if any, could be strengthened to contribute to diabetic vascular complications. Given the broad expression of all these proteins in the nanocomplex in different tissues/cells, we speculate that results here may have braod impact across many fields.

AKAP5 ablation disrupted most protein interactions in the nanocomplex except for P2Y$_{11}$ and AC5, which maintained similar levels of association in WT and AKAP5$^{-/-}$ cells. These results were not entirely surprising as a recent study has suggested that GPCRs can exist in a pre-coupled state with ACs[56]. Despite this, we submit that the multifaceted genetic and nanoscale imaging approaches presented herein strongly supports that AKAP5, P2Y$_{11}$, AC5, PKA, and Ca$_V$1.2 can form part of a nanomolecular complex in arterial myocytes. Consequently, this complex is positioned to regulate vascular L-type Ca$^{2+}$ channel activity and vascular reactivity upon elevated glucose. Indeed, altering complex function by knocking out AC5 and AKAP5 or pharmacological blocking of P2Y$_{11}$/P2Y$_{11}$-like receptors prevented potentiation of L-type Ca$^{2+}$ channels and vasoconstriction in response to elevated glucose or NF546 in ex vivo and in vivo preparations (Figs. 3 and 4)[20]. These results correlated with a reduction in blood flow in response to elevated glucose and NF546 in WT but not AKAP5$^{-/-}$ mice (Fig. 4). One therapeutic implication of these findings may be that inhibition of the P2Y$_{11}$ or disruption of the AKAP5/GPCR or GPCR/AC association could be developed into a pharmacological strategy for treating diabetic vascular complications. Future studies should investigate whether the administration of a P2Y$_{11}$ inhibitor or disruptor of the AKAP/P2Y$_{11}$ or P2Y$_{11}$/AC5 interface can prevent/ameliorate altered vascular reactivity and vascular complications in diabetes.

The preserved association between P2Y$_{11}$ and AC5 in AKAP5$^{-/-}$ arterial myocytes raises an important question: why glucose and the P2Y$_{11}$ agonist NF546 are not able to stimulate cAMP synthesis in AKAP5$^{-/-}$ cells, and for that matter in cells in which AKAP function is altered with the ht31 peptide? It is possible that disruption of AKAP function may alter PDEs compartmentalization,

activity or key feedback regulatory loops to prevent glucose/NF546-induced cAMP synthesis[25,54,55,57]. Consistent with this possibility, it was found that glucose and NF546 triggered similar magnitudes of cAMP production in WT and AKAP5$^{-/-}$ arterial myocytes pre-treated with IBMX (Fig. 7a, b and Supplementary Fig. 10). These data suggest that P2Y$_{11}$/P2Y$_{11}$-like receptors are still able to trigger downstream cAMP synthesis in the absence of AKAP5, but this cAMP production is potentially prevented/masked by an increase in PDE activity or its redistribution. Intriguingly, despite the rescue of the glucose/NF546-induced cAMP signal in IBMX-pretreated AKAP5$^{-/-}$ cells, this was not sufficient to induce constriction in response to the same stimuli in these cells. These results indicate that AKAP5 is still fundamentally required for glucose/NF546-induced vasoconstriction, likely by maintaining P2Y$_{11}$/P2Y$_{11}$-like receptors, AC5, PKA, and Ca$_V$1.2 in nanometer proximity of each other to tune L-type Ca$^{2+}$ channel activity, intracellular Ca$^{2+}$ concentration and the contractile machinery. Future studies should define the mechanisms by which AKAPs, in particular AKAP5, control PDE activity to modulate cAMP signaling and excitability in arterial myocytes.

AKAP5 also modulates transcriptional remodeling upon chronic hyperglycemic conditions[46,47]. This may proceed through the complex array of the other kinase anchoring functions of AKAP5[58,59]. For example, in arterial myocytes, AKAP5 mediates PKA and PKC-dependent regulation of L-type Ca$^{2+}$ channels[10–12,32,47]. The same protein–protein interactions may also contribute to activate the phosphatase calcineurin and the transcription factor NFATc3 upon chronic hyperglycemia and diabetes[10,46,47,60]. Notably, the role of AKAP5 in targeting calcineurin to Ca$_V$1.2 facilitates the Ca$^{2+}$/calmodulin-dependent activation of the phosphatase in response to increased Ca$^{2+}$ influx via L-type Ca$^{2+}$ channels during chronic hyperglycemia and diabetes[47,60]. The activated calcineurin can then dephosphorylate NFATc3 leading to an increase in its nuclear accumulation that promotes transcriptional remodeling. Indeed, activation of this pathway plays a central role in the selective downregulation in the expression and function of the BK$_{Ca}$ β1 and K$_V$2.1 subunits upon chronic hyperglycemia and in diabetes[35,46,47]. The physiological consequences of this AKAP5-anchored calcineurin-mediated reduction in BK$_{Ca}$ β1 and K$_V$2.1 function are an enhanced myogenic tone due to a decrease in the negative feedback regulation of the membrane potential of arterial myocytes[12,46,47,61]. Thus, AKAP5 could influence membrane potential-dependent and independent mechanisms to control arterial myocyte contractility upon chronic hyperglycemia. Whether AKAP5 is still necessary for activation of the calcineurin/NFATc3 signaling pathway in response to direct stimulation of the P2Y$_{11}$ receptor remains the subject for future studies. Such studies could provide additional insight into the contributions of the AKAP5/P2Y$_{11}$/AC5/PKA/Ca$_V$1.2 nanocomplex in the development of vascular complications during chronic hyperglycemia and diabetes.

Other AKAPs could potentially interact with P2Y$_{11}$, AC5, PKA, and Ca$_V$1.2 to promote the formation of this nanomolecular complex. In cardiomyocytes, both AKAP5 and AKAP7 (also referred to as AKAP15 or AKAP18) have been shown to interact with β1 adrenergic receptors, AC, PKA, and Ca$_V$1.2 to form macromolecular complexes[30,62]. Our results, however, suggest a central role for AKAP5 in orchestrating the P2Y$_{11}$/AC5/PKA/Ca$_V$1.2 nanocomplex to regulate cAMP production, L-type Ca$^{2+}$ channel activity, vascular reactivity, and blood flow in response to elevated glucose and NF546. Thus, whether other AKAPs are necessary for modulation of the aforementioned signaling pathway by glucose or P2Y$_{11}$ agonists is unlikely.

Increases in extracellular glucose are known to induce the autocrine release of nucleotides in arterial myocytes[20,21,44]. The release of nucleotides activates purinergic receptors to modulate intracellular Ca$^{2+}$ and NFATc3 signaling in arterial myocytes[20,44]. Nucleotides can be transported out of the cell via multiple pathways, including nucleotide-binding cassettes, connexin hemichannels, and/or pannexin channels[63]. It is intriguing to speculate that either of these proteins could interact with AKAP5 in the nanocomplex to facilitate nucleotide signaling to the P2Y$_{11}$/AC5/PKA/Ca$_V$1.2 pathway.

In summary, our data reveal that AKAP5 sustains a nanocomplex composed of P2Y$_{11}$/P2Y$_{11}$-like receptors, AC5, PKA, and Ca$_V$1.2 in arterial myocytes. This AKAP5-driven nanocomplex affords the necessary localization of P2Y$_{11}$/P2Y$_{11}$-like receptors within the vicinity of subpopulations of AC5, PKA, and Ca$_V$1.2 to coordinate compartmentalized signaling in response to elevations in extracellular glucose. The glucose/NF546-induced spatial confinement of P2Y$_{11}$/P2Y$_{11}$-like/AC5-mediated cAMP is essential to facilitate activation of distinct pools of PKA that stimulate a subpopulation of L-type Ca$^{2+}$ channels to increase global [Ca$^{2+}$]$_i$ leading to vasoconstriction during hyperglycemia. The clinical implications of this nanocomplex are significant as they shed light on a mechanism underlying altered vascular reactivity during diabetic hyperglycemia.

## Methods

**Key resources**. Key resources are documented in Supplementary Table 5.

**Animals**. All experiments were performed in strict compliance with the University of California Davis ethical regulations for studies involving animals as approved by the University of California Davis Animal Care and Use Committee (protocols #: 20321 and 20234). Age-matched (5–8 weeks) male C57BL/6J (WT), AKAP5$^{-/-}$, and AC5$^{-/-}$ mice were euthanized with a lethal dose of pentobarbital (250 mg/kg intraperitoneally). The AKAP5$^{-/-}$ and AC5$^{-/-}$ have been backcrossed into the C57BL/6J background for 10 generations[19,30,32,48]. For some electrophysiology experiments, age-matched (5–8 weeks) female C57BL/6J (WT) and AKAP5$^{-/-}$ mice were used.

**Human tissue**. Excised adipose arteries from obese, nondiabetic human male and female patients undergoing surgical sleeve gastrectomy were used for this study (see Supplementary Table 1). Samples were obtained in accordance with the guidelines of the *Declaration of Helsinki* and after Institutional Review Board (IRB) approval from the University of Nevada Reno School of Medicine (IRB ID: 2013-019). Since the tissue would have been otherwise disposed, has no codification that could be used to identify patients and was determined not to be human subject research in accordance with United States of America federal regulations as defined by 45 CFR 46.102(f), the need for informed consent was waived by IRBs at the University of Nevada Reno School of Medicine (IRB ID: 2013-019) and the University of California Davis (IRB ID: 597267-1). This precludes the acquisition of detailed clinical profiles other than sex, age and whether the patient had diabetes or not. Thus, no exclusions were made due to medication history or presence of comorbidities. Collected tissue was placed in cold phosphate-buffered saline (PBS) solution containing (in mM): 138 NaCl, 3 KCl, 10 Na$_2$HPO$_4$, 2 NaH$_2$PO$_4$, 5 D-glucose, 0.1 CaCl$_2$, and 0.1 MgSO$_4$, pH 7.4, with NaOH until use.

**Arterial myocyte isolation**. Mouse cerebral arteries were dissected in ice-cold dissection buffer containing (in mM): 140 NaCl, 5 KCl, 2 MgCl$_2$, 10 D-glucose, and 10 HEPES, pH 7.4, with NaOH. Arteries were digested in dissection buffer containing papain (1 mg/mL) and dithiothreitol (1 mg/mL) at 37 °C for 7 min, followed by incubation in dissection buffer containing collagenase type F (0.7 mg/mL) and collagenase type H (0.3 mg/mL) at 37 °C for 7 min. Arteries were washed in ice-cold dissection buffer and then gently triturated with glass pipettes to disperse the cells. Arterial myocytes were kept in ice-cold dissection buffer until use.

Single human arterial myocytes were isolated from small diameter adipose arteries from human samples. The dissected human arteries were enzymatically digested in dissection buffer (as above) containing papain (26 U/mL) and dithiothreitol (1 mg/mL) at 37 °C for 15 min, followed by incubation in dissection buffer supplemented with collagenase type H (1.95 U/mL), elastase (0.5 mg/mL), and trypsin inhibitor (1 mg/mL) at 37 °C for 10 min[12,20,35]. Arteries were washed three times in ice-cold dissection buffer and triturated with glass pipettes to disperse individual cells, which were maintained in ice-cold dissection buffer until use.

For unpassaged, cultured human and mouse arterial myocytes, human adipose arteries, and mouse aortas were dissected out. The adventitia was removed using forceps. Subsequently, arteries were cut open to remove the endothelial layer using cotton swabs, and tissue was placed in ice-cold Dulbecco's modified Eagle medium

(DMEM; Gibco – Life Technologies, Grand Island, NY) containing 1X glutamate, 1X pyruvate, 1X penicillin/streptavidin, and fungizone (0.25 g/mL)[19,20]. Artery segments were transferred and incubated in a DMEM solution containing 2.2 mg/mL of collagenase Type 2 (Worthington) at 37 °C for 15 min. To disperse and culture unpassaged arterial myocytes, the tissue was cut into 2–5 mm segments and incubated at 37 °C with constant shaking in a buffer containing (in mM): 134 NaCl, 6 KCl, CaCl$_2$, 10 HEPES, and 7 D-glucose supplemented with 2.2 mg/mL collagenase Type 2 (Worthington). The digestion was stopped after adding an equal volume of DMEM containing 5% fetal bovine serum. Digested tissue was centrifuged for 5 min at $30,100 \times g$. The supernatant was removed, and the pellet containing the digested tissue was resuspended in DMEM containing 1X glutamate, 1X pyruvate, 5% serum, and 5 mM D-glucose, allowing dispersion of individual arterial myocytes. Cells were then seeded on glass coverslips coated with laminin and kept in an incubator at 37 °C with 5% CO$_2$ for 2–3 days before adenoviral transduction.

**Flow cytometry.** For flow cytometry, single arterial myocytes from 2 to 3 days unpassaged cultures, obtained as described in the arterial myocyte isolation section, were filtered through a 200 μm cell strainer and fixed in 0.4% paraformaldehyde. Cells were treated with Alexa Fluor 488-conjugated anti-α smooth muscle actin (Abcam, Cambridge, MA), phytoerythrin-conjugated anti-Thy1.2 (BD Bioscience, San Diego, CA), anti-CD31 (BD Bioscience), anti-CD45 (BD Bioscience, San Diego, CA) antibodies and lineage antibody cocktail (CD3e, CD11b, Cd45R, Ly-6C, Ly-6G, and TER-119, BD Bioscience, San Diego, CA) in PBS with 5% donkey serum and 20 μg/mL DNASe-free RNAse (Sigma) overnight at 4 °C. An additional staining was made with 40 μg/mL 7-aminoactinomycin D (7AAD, BD Bioscience, San Jose, CA) to measure the DNA content. Data were collected using a standard FACScan cytometer (BD Biosciences, San Jose, CA) upgraded to a dual laser system with the addition of a blue laser (15 mW at 488 nm) and a red laser (25 mW at 637 nm Cytek Development, Inc, Fremont, CA). Data were acquired and analyzed using CellQuest v5.2.1 (BD Bioscience, San Diego, CA) and FlowJo software (ver9.9.6 Treestar Inc., San Carlos, CA), respectively.

**Adenovirus infection of arterial myocytes and FRET.** Laminin (Life Technologies, Grand Island, NY) diluted 100× in sterile-filtered PBS (137 mM NaCl, 2.7 mM KCl, 10 mM Na$_2$HPO$_4$, 1.8 mM KH$_2$PO$_4$, pH = 7.4) was used to coat #0 glass coverslips (Karl Hecht, Sondheim, Germany). Diluted laminin (100 μL) was added to each coverslip, and coverslips were placed in a 37 °C incubator with 5% CO$_2$ for a minimum of 2 h, then moved to 24-well plate wells (Falcon, Tewksbury, MA). Coverslips were washed three times with sterile-filtered PBS. Freshly dissociated human adipose arterial myocytes and mouse aortic cells, obtained as described in the arterial myocyte isolation section, were plated on the laminin-coated coverslips with 500 μL of serum-containing media for 2–3 days in a 37 °C incubator with 5% CO$_2$. Following this incubation period, media was then replaced with 500 μL of serum-free media-containing virus coding for the membrane-targeted Epac1-camps-based FRET sensor (ICUE3-PM)[33,64] and incubated at 37 °C with 5% CO$_2$ for another 36 h. Viruses were produced using the AdEasy system (Qbiogene, Carlsbad, CA)[65]. Media was changed to serum-free media without virus after infection. Glass coverslips were transferred to glass-bottom culture dishes (MatTek, Ashland, MA) containing 3 mL PBS for experiments.

Phase contrast, CFP480, and FRET images were acquired with a Zeiss AXIO Observer A1 inverted fluorescence microscope (San Diego, CA) equipped with a Hamamatsu Orca-Flash 4.0 digital camera (Bridgewater, NJ) and controlled by Metaflor software v7.7 (Molecular Devices, Sunnyvale, CA). FRET images were collected using only the ×40 oil-immersion objective lens, whereas phase contrast and CFP480 images were collected with ×20 and ×40 oil-immersion objective lenses. To acquire images for FRET analysis, the donor fluorophore was excited at 430–455 nm and emission fluorescence was measured with two filters (475DF40 for cyan and 535DF25 for yellow). Images were subjected to background subtraction and acquired every 30 s with exposure time of 200 ms for each channel. The donor/acceptor FRET ratio was calculated and normalized to the ratio value of baseline. The donor fluorophore was excited at 430–455 nm and emission fluorescence was measured with the 475DF40 filter for 25 ms to acquire CFP480 images. Averages of normalized curves and maximal response to stimulation were graphed based on FRET ratio changes before treatment. Increases in cAMP levels were represented by decreases in the YFP/CFP ratio that occur when cAMP binds to the ICUE3-PM sensor. Experiments were performed at room temperature (22–25 °C).

**Immunoblotting.** Mouse cerebral and mesenteric arteries, were homogenized in a RIPA lysis buffer solution (mM): 50 Tris base, 150 NaCl, 5 EGTA, 10 EDTA, 1% nonyl phenoxypolyethoxylethanol-40 (NP-40), 10% glycerol, 0.05% sodium dodecyl sulfate (SDS), 0.4% deoxycholic acid (DOC) with protease inhibitors (1 μg/mL pepstatin A, 10 μg/mL leupeptin, 20 μg/mL aprotinin, and 200 nM phenylmethylsulfonyl fluoride). The lysate was then cleared by centrifugation (250,000 × g, 30 min, 4 °C). The resulting supernatant was used as the whole arterial lysate. Samples were boiled in Laemmli Sample Buffer (Bio-Rad) for 5 min at 95 °C. Proteins were separated by SDS-polyacrylamide gel electrophoresis (75–100 V; 1.5 h) in a separating phase polymerized from 3% acrylamide and a resolving phase polymerized from 7.5% acrylamide. Separated proteins were then transferred to a polyvinylidene difluoride membrane (50 V, 600 min, 4 °C). Membranes were blocked

in either 5% nonfat dried milk in tris-buffered saline with 0.05% Tween 20 (TBS-T) (AC5) or 10% Odyssey blocking buffer (P2Y$_{11}$) for 1 h at room temperature before incubation in primary antibody for 3 h at room temperature. The P2Y$_{11}$ antibody was diluted in 1% bovine serum albumin in TBS-T (1:100-1:200 goat anti-P2Y$_{11}$ sc69588 from Santa Cruz). The AC5 antibody was diluted in 5% nonfat dried milk in TBS-T (1:500 goat anti-AC5 sc74301 from Santa Cruz Biotechnology). This was followed by incubation of membrane in horseradish peroxidase-labeled donkey anti-goat antibodies (1:10,000 from Bio-Rad) in 5% nonfat dried milk TBS-T. Classico (Millipore) and Femto (Thermo) chemiluminescence reagents and exposure to X-ray film were used to identify the bands. The PKA (1:500 mouse anti-PKA RIIα 612242 from BD Bioscience) and Ca$_V$1.2 (9.6 μg/ml of FP1 rabbit anti-Ca$_V$1.2[66]) antibody was diluted in 5% nonfat dried milk TBS-T. Blots were incubated with primary antibodies overnight, followed by incubation with secondary antibodies in 5% nonfat dried milk TBS-T with IRDye 800CW goat anti-mouse/rabbit (1:5000) (Abcam). Antibody binding was detected using a Chemidoc MP system (Bio-Rad), where secondary antibodies were excited at 776 nm and emission collected at 792–900 nm. Densitometry analysis for bands was performed with ImageJ software v1.51 (National Institutes of Health). Total protein was used for quantification of relative protein density in WT and AKAP5$^{-/-}$ arterial lysates.

**Immunofluorescence.** For immunofluorescent labeling of freshly dissociated arterial myocytes[8,20,32,35], cells were plated and allowed to adhere to a coverslip for 1 h. Cells were washed with PBS (1× for 15 min), fixed with 3% glyoxal (20 min), quenched with 100 mM glycine (15 min), and washed with PBS (2× for 3 min). Cells were permeabilized with 0.1% Triton X-100 (20 min), blocked in 50% Odyssey blocking solution (LI-COR Bioscience) for 1 h at 37 °C, and incubated overnight at 4 °C. The primary antibodies were diluted in a solution consisting of 0.1% Odyssey + 0.05% Triton X-100 in PBS. Simultaneous incubation of AKAP5/P2Y$_{11}$/Ca$_V$1.2 and AKAP5/AC5/Ca$_V$1.2 antibodies proceeded overnight at 4 °C (10 μg/mL rabbit anti-AKAP5 Millipore 07-210, 10 μg/mL goat anti-AC5 Santa Cruz Biotechnology sc74301, and 10 μg/mL mouse anti-Ca$_V$1.2 Neuromab clone N263/31). The following day, cells were rinsed three times and washed with PBS (3× for 15 min). The secondary antibodies were Alexa Fluor 430-conjugated goat anti-rabbit, Alexa Fluor 568-conjugated donkey anti-mouse and Alexa Fluor 647-conjugated donkey anti-goat (5 mg/mL; Molecular Probes). Secondary antibodies were diluted in PBS buffer. First, cells were incubated with the Alexa Fluor 647-conjugated donkey anti-goat (5 mg/mL; Molecular Probes) for 1 h at room temperature followed by a rinse with PBS (3× for 10 min). Cells were then incubated with the remaining secondary antibodies (Alexa Fluor 430-conjugated goat anti-rabbit and Alexa Fluor 568 donkey anti-mouse; 5 mg/mL; Molecular Probes) for 1 h at room temperature after which, cells were washed with PBS (3× for 10 min). In control experiments, cells were incubated with 10 μg/mL normal rabbit IgG (Cell Signaling 2729S), 10 μg/mL normal goat IgG (Millipore NI02), and 10 μg/mL normal mouse IgG (Millipore NI03) primary antibodies followed by Alexa Fluor secondary antibodies as described above. Cells were sequentially imaged using a Zeiss 880 confocal laser-scanning microscope equipped with an Airyscan detector module, a Plan-Apo 63× 1.4 NA oil-immersion objective and 405/561/647 lasers. Cells for each group were imaged using the same acquisition parameters. Raw images were subsequently processed using the Zen software v2.3 SP1 to achieve a sub-diffraction-limited resolution image (~120 μm). Images were background subtracted, pseudo-colored and analyzed offline using ImageJ v1.51.

**GSD super-resolution microscopy.** Cells were plated and allowed to adhere to a coverslip for 1 h. Cells were washed three times with PBS for 15 min, fixed with 3% paraformaldehyde + 0.1% glutaraldehyde solution in PBS (10 min), and incubated for 5 min in 0.1% sodium borohydride. After washing with PBS (3× for 20 min), the fixed cells were permeabilized and blocked with 0.05% Triton X-100 and 20% SEA BLOCK (Thermo Scientific) in a PBS solution for 1 h at room temperature. Primary antibodies diluted in the blocking solution were added and kept overnight at 4 °C (10 μg/mL rabbit anti-AKAP5 Millipore 07-210 or goat anti-AKAP5 Santa Cruz Biotechnology clone C-20, 10 μg/mL goat anti-P2Y$_{11}$ Santa Cruz Biotechnology clone C-18, 10 μg/mL goat anti-AC5 Santa Cruz Biotechnology sc74301, and 10 μg/mL custom anti-FP1[66]). Following 12 h, cells were washed three times with PBS for 20 min. Secondary antibodies included Alexa Fluor 647-conjugated donkey anti-goat (2 μg/mL; Molecular Probes) and Alexa Fluor 568-conjugated donkey anti-rabbit (2 μg/mL; Molecular Probes). Secondary antibodies were diluted in blocking buffer, cells were incubated in them for 1 h at room temperature and subsequently washed three times with PBS (20 min). Specificity of secondary antibodies was tested in control experiments in which primary antibodies were omitted from the preparation (no 1° antibody control).

Imaging was performed using coverslips mounted on a round cavity microscope slide containing MEA-GLOX imaging buffer (NeoLab Migge Laborbedarf-Vertriebs GmbH, Germany) and sealed with Twinsil (Picodent, Germany). Imaging buffer composition contained 10 mM MEA (cysteamine), 0.56 mg/mL glucose oxidase, 34 μg/mL catalase, and buffer containing 10% w/v glucose, 10 mM NaCl, and 50 mM Tris-HCl, pH 8. Images were obtained using a super-resolution ground state depletion system (SR-GSD, Leica) dependent on stochastic single-molecule localization and equipped with high-power lasers (532 nm 2.1 kW; 642 nm 2.1 kW; 405 nm 30 mW). A 160× HCX Plan-Apochromat (NA 1.47) oil-immersion lens and an electron-multiplying charge-coupled device (EMCCD)

camera (iXon3 897; Andor Technology) were used to acquire images. The camera was running in frame-transfer mode at a frame rate of 100 Hz (10 ms exposure time). Fluorescence was detected through Leica high-power TIRF filter cubes (532 HP-T, 642 HP-T) with emission band-pass filters of 550–650 and 660–760 nm.

AKAP5, P2Y11, AC5, and Cav1.2 distributions were reconstructed from 30,000 images using the coordinates of centroids obtained by fitting single-molecule fluorescence signals with a 2D Gaussian function in LASAF software (Leica). The localization accuracy of the system is limited by the statistical noise of photon counting. The precision of localization is proportional to $DLR/\sqrt{N}$, where DLR is the diffraction-limited resolution of a fluorophore and $N$ is the average number of detected photons per switching event, assuming the point-spread functions are Gaussian[67,68]. The full width at half maximum for single-molecule signals was ~20–40 nm[51,69]. Localizations produced from <800 photons were filtered out of the reconstruction.

All pixels with intensity above a user-defined threshold were binarized and segmented into individual objects and included as clusters in our analysis. Cluster size and density were determined using the Analyze Particle option in the ImageJ v1.51 software (National Institute of Health). The JACoP plug-in in the ImageJ v1.51 software was used to automatically determine the shortest intermolecular distances for AKAP5-P2Y11, AKAP5-AC5, and AKAP5-Cav1.2. Intermolecular distance histograms were generated from the JACoP plug-in output data and fitted with a sum of two Gaussian functions included in Origin v7.0 software. To calculate the percentage of AKAP5 overlap, the thresholded super-resolution localization map images for AKAP5-P2Y11, AKAP5-AC5, and AKAP5-Cav1.2 were multiplied in ImageJ v1.51. Clusters were calculated from the resultant image and compared with the total number of objects in the AKAP5 super-resolution localization map. Random simulation of image pairs for AKAP5-P2Y11, AKAP5-AC5, and AKAP5-Cav1.2 was generated using the Coste's randomization algorithm in ImageJ v1.51[52]. Images were analyzed as described above.

**Proximity ligation assay.** A Duolink In Situ PLA kit (Sigma)[53] was used to detect complexes consisting of P2Y11 and AKAP5 (i.e., AKAP79 in humans and AKAP150 in rodents), P2Y11 and AC5, P2Y11 and Cav1.2, P2Y11 and PKAcat, AC5 and AKAP5, Cav1.2 and AKAP5, and Cav1.2 and PKAcat/PKARIIα in freshly isolated arterial myocytes[12,19,20,35]. Freshly isolated cells were plated and allowed to adhere on glass coverslips (1 h, room temperature), followed by fixation with 3% glyoxal (20 min)[70], quenching with 100 mM glycine (15 min), and PBS washes (2× for 3 min). Cells were then permeabilized with 0.1% Triton X-100 (20 min), blocked in 50% Odyssey blocking solution (LI-COR Bioscience) for 1 h at 37 °C, and incubated overnight at 4 °C with a specific combination of two primary antibodies in 0.1% Odyssey + 0.05% Triton X-100 PBS solution: goat anti-P2Y11 (1:100; Santa Cruz Biotechnology, clone C-18), rabbit anti-P2Y11 (1:100; Abcam, ab180739), rabbit anti-AKAP79 (1:200; Millipore ABS102), rabbit anti-AKAP5 (1:200; Millipore 07-210), goat anti-AKAP5 (1:200; Santa Cruz Biotechnology, clone C-20), goat anti-AC5 (1:1000; Santa Cruz Biotechnology, sc74301), rabbit anti-PKAcat α, β, γ (1:200; Santa Cruz Biotechnology, clone H-95), mouse anti-PKARIIα (1:50; BD Transduction Laboratories 612242), custom rabbit anti-FP1 (1:100; human dataset)[66], and mouse anti-Cav1.2 (1:200; Neuromab, clone N263/31; human dataset). Cells incubated with only one primary antibody or with normal rabbit IgG (1:200; Cell Signaling 2729S) and normal goat IgG (1:200; Millipore NI02) were used as negative controls. After primary antibody incubation, cells were washed with Duolink buffer A (2× for 5 min). Oligonucleotide-conjugated secondary antibodies (PLA probes: anti-goat MINUS, anti-mouse MINUS, and anti-rabbit PLUS) were used to detect P2Y11 + AKAP5, P2Y11 + AC5, P2Y11 + Cav1.2, P2Y11 + PKAcat, AC5 + AKAP5, Cav1.2 + AKAP5, Cav1.2 + PKAcat (human) or PKARIIα (mouse), and rabbit IgG + goat IgG (1 h, 37 °C). Ligation and amplification steps were carried out according to the manufacturer instructions. Coverslips were allowed to dry and subsequently mounted on a microscope slide with Duolink mounting medium. An Olympus FV1000 confocal microscope coupled with a ×60 oil-immersion lens (NA, 1.4) was used to visualize fluorescence signal. Images were acquired at different optical planes (z-axis step = 0.5 μm) using the Olympus Fluoview v1.4 software. For each sample, a single-intensity projection image from the combined stack of images was used for analysis of the number of puncta per cell area (μm²)[12,19,20,35]. Analysis was performed using the NIH ImageJ open software v1.51. These data were analyzed by members of the team that were blind to conditions.

**Electrophysiology.** All experiments were performed at room temperature (22–25 °C) and acquired using an Axopatch 200B amplifier and Digidata 1440 digitizer (Molecular Devices). Recording electrodes pulled from borosilicate capillary glass using a micropipette puller (model P-97, Sutter Instruments) were polished to achieve resistances that ranged from 3.5 to 5.0 MΩ. L-type channel currents were assessed in freshly dissociated arterial myocytes using the perforated whole-cell mode of the patch-clamp technique with $Ba^{2+}$ as a charge carrier ($I_{Ba}$) after myocytes were allowed to attach (10 min) to a glass coverslip in a recording chamber. Borosilicate glass pipettes were filled with a solution consisting of (in mM): 120 CsCl, 20 tetraethylammonium chloride (TEA-Cl), 1 EGTA, and 20 HEPES with amphotericin B (250 μg/mL; pH adjusted to 7.2 with CsOH). The bath solution consisted of (in mM): 115 NaCl, 10 TEA-Cl, 0.5 $MgCl_2$, 10 D-glucose, 5 CsCl, 20 $BaCl_2$, and 20 HEPES, pH adjusted to 7.4 with CsOH. $I_{Ba}$ were evoked by 200 ms depolarizing pulses from a holding potential of −70 to +10 mV. Currents were sampled at 10 kHz, low-pass-filtered at 5 kHz and recorded using Clampex v10.3 (Molecular Devices). The Cav1.2 blocker nifedipine (1 μM) was applied at the end of every experiment to determine the nifedipine-sensitive component. A voltage error of 10 mV attributable to the liquid junction potential of the recording solutions was corrected offline. Currents were analyzed using Clampfit v10.3 (Molecular Devices).

**Arterial diameter measurements.** Freshly isolated posterior mouse cerebral arteries were cannulated on glass micropipettes mounted in a 5 mL myograph chamber[12,19,20,35,46]. Arteries were equilibrated at an intravascular pressure of 20 mmHg and continuously perfused (37 °C, 30 min, 3–5 mL/min) with artificial cerebral spinal fluid (aCSF) solution containing (in mM): 119 NaCl, 4.7 KCl, 2 $CaCl_2$, 24 $NaHCO_3$, 1.2 $KH_2PO_4$, 1.2 $MgSO_4$, 0.023 EDTA, and 10 D-glucose. Bath solution was aerated with 5% $CO_2$/95% $O_2$ to maintain pH at 7.35–7.40. Diameter recordings were performed using an IonOptix system, and analysis of the diameter data was performed using the IonOptix v6.6 software. After equilibration, arteries were challenged with 60 mM $K^+$ to test viability. Arteries with a robust response to 60 mM $K^+$ were pressurized to 60 mmHg to allow development of stable myogenic tone. Arteries that did not exhibit stable tone after ~1 h were discarded. Once the arteries had achieved stable tone, vessels were submitted to the different treatment conditions, as described throughout the main text. Myogenic tone data are presented as a percent decrease in diameter relative to the maximum passive diameter at 60 mmHg obtained using $Ca^{2+}$-free saline solution containing 1 μM nifedipine (0 $Ca^{2+}$/nif) at the end of each experiment.

**In vivo intravital imaging and blood flow measurements.** Mice were anesthetized with isoflurane (5% induction and 2% maintenance for the duration of the surgery). Once mice reached stage III anesthesia, the skin above the skull was removed to expose the top of the cranium. Dental cement and superglue were used to attach a steel plate with ~2 mm diameter hole on the region of interest. After securing the head plate to a holding frame, a circular cranial window ~2 mm in diameter was drilled in the skull. Mice were injected with ketamine (100 mg/kg) and xylazine (10 mg/kg) to replace 2% isoflurane-induced anesthesia. An electrical heating pad was used to maintain body temperature at 37 °C. In vivo intravital imaging of pial arteries was done using a Unitron Z850 stereomicroscope and a StCamSware software v3.10.0.3086 (Sentech America Incorporated). Blood flow was measured in pial arteries using a Moor FLPI-2 Laser Speckle Contrast Imager (Moor Instruments). Different treatments were acutely added to the cranial window in physiological saline solution composed of (in mM): 119 NaCl, 4.7 KCl, 2 $CaCl_2$, 24 $NaHCO_3$, 1.2 $KH_2PO_4$, 1.2 $MgSO_4$, 0.023 EDTA, and 10 D-glucose, aerated with 5% $CO_2$/95% $O_2$ (pH 7.35–7.4, 37 °C). Passive diameter was obtained after application of 0 $Ca^{2+}$ + 1 μM nifedipine (0 $Ca^{2+}$/nif). Arterial diameter was analyzed using NIH ImageJ v1.51 Line Measurement tool by drawing a line across the arterial walls before and after treatments. The percentage myogenic tone was calculated using the following equation:

$$Myogenic\ tone = \left[\frac{(DP - DA)}{DP}\right] \times 100 \qquad (1)$$

where DP = passive (in 0 $Ca^{2+}$/nif aCSF solution) diameter and DA = active (in $Ca^{2+}$ containing aCSF solution) diameter. Blood flow was quantified using the Moor FLPI Review v5.0 software and values were normalized to blood flow measured in response to 0 $Ca^{2+}$ + 1 μM nifedipine (0 $Ca^{2+}$/nif). These data were analyzed by members of the team that were blind to conditions.

**Chemicals.** All chemical reagents were from Sigma-Aldrich (St. Louis, MO) unless otherwise stated. A list of key reagents/resources is included in Supplementary Table 5.

**Statistics.** Data were analyzed using GraphPad Prism v6.0 or Origin v7.0, and expressed as mean ± SEM. All the experimental series used technical replicates, as they examined acute effects. The number of technical replicates is based on previously published observations to reached statistical differences between datasets[11,12,20,32,35,46,47], which is consistent with the general number of replicates and conditions commonly accepted in the field. To account for potential variability in sample preparation, animals, ambiance conditions and other factors, datasets were produced from cells obtained from at least 2 different mice/human preparations. Number of technical and biological replicates are included in figure legends. All attempts at replication were successful. No data were excluded. Data were assessed for potential outliers using the GraphPad Prism Outlier Test and for normality of distribution. Statistical significance was then determined using appropriate paired or unpaired two-tailed Student's t-test, nonparametric tests or one-way analysis of variance (ANOVA) for multiple comparisons with appropriate post hoc test. $P < 0.05$ was considered statistically significant (denoted by * in figures).

**Reporting summary**. Further information on research design is available in the Nature Research Reporting Summary linked to this article.

## Data availability

All data generated or analyzed in this study are included in the main manuscript and/or supplementary figures. Raw data of images is available upon request. Source data are provided with this paper.

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

## Acknowledgements

We thank members of the Navedo Lab for technical support and for critically reading early versions of the manuscript. We thank Moor Instruments for kindly loaning the Moor FLPI-2 Laser Speckle Contrast Imager, and Dr. Yi-Je Chen in the Department of Pharmacology at the University of California Davis for organizing the demo of this equipment. This work was supported by NIH grants R01HL098200, R01HL121059, and R01HL149127, and UC MEXUS-CONACYT CN-19-147 (to M.F.N.), T32GM099608 (to M.P.P.), T32HL086350 (to A.U.S.), R01GM127513 (to E.J.D.), R01AG055357 (to J.W.H.), R01DK105542 and DK119192 (to J.D.S.), R01HL127764 and R01HL112413 (to Y.K.X.), and a UC Davis Academic Federation Innovative Development Award (to M.N.-C.). M.N.-C. is a UC Davis CAMPOS Fellow.

## Author contributions

Our multiscale approach required different expertise, which is reflected in the sharing of the first author position. M.P.P. took a leading role in collecting and organizing the figures provided by all authors and wrote the first version of the manuscript. M.P.P., A.U.S., and G.R.: designed and executed experiments, validated protocols, collected, analyzed and interpreted data, and revised the manuscript. M.M.-A.B., V.A.F.-T., P.S., and P.B.: executed experiments and revised the manuscript. K.C.S. and S.M.W.: provided human samples and revised the manuscript. N.C., E.J.D., J.W.H., J.D.S., L.F.S., and Y.K.X.: provided reagents and access to equipment and revised the manuscript. M.F.N. and M.N.-C.: conceived the entire project and designed experiments, collected, analyzed and interpreted data, wrote and revised the manuscript, and provided overall super-vision and direction to the project.

## Competing interests

The authors declare no competing interests.
