## [Peer Review File · Nature Communications]

Reviewers' comments:

Reviewer #1 (Remarks to the Author):

This very exciting paper focuses on the role of the gene product of AKAP5 in Gs-coupled purinergic modulation of CaV1.2 channels in vascular smooth muscle cells, and its subsequent vasoconstriction. This concept is similar to, and builds on, similar reports for most members of this team on cardiac myocytes and neurons in brain. The physiological process investigated revolves around how extracellular glucose potentiates the activity of vascular CaV1.2 channels. It has been known for some years that this signaling mechanism involves PKA, AC5 and Gs-coupled P2Y11 receptors. The central advance of this manuscript is the orchestration of this signaling pathway by AKAP79/150 by grouping these signaling elements into "nanodomains," a role this scaffold protein has been shown to play in brain and muscle for over 5 years. The main approaches used to support this hypothesis include pharmacology, AKAP5 KO mice, FRET AKAP5-mediated PKA reporters, vascular tone and flow measurements and PLA assays.

Critique:

1. Most of the elements of this signaling pathway have been previously described by about a dozen publications, mostly by this same group, which I don't need to delineate here. Thus, the signaling pathway involved as described here is not novel. What is novel is the role of AKAP5 gene products in clustering these proteins into "nanodomains" in intimate proximity. I believe this hypothesis is likely to be true, however, the data set in this manuscript do not establish it firmly.
2. Although I appreciate the PLA assays as indicative of this hypothesis, as well as the results from AKAP5 KO mice, I think what is lacking are super-resolution imaging of such complexes as has been performed robustly, and well, by members of this team. Indeed, I was surprised that such experiments were not included in this manuscript. I know this team is capable of performing such experiments in an expert way, and I think for this journal, such data should be included. I do not think that the PLA assays presented are sufficient.
3. Minor. I am not fond of the presentation style of the figures. The black bars and traces superimposed on the light pink bars and traces are hard to discern. In addition, there are other issues in the presentation of the data, such as the perhaps superfluous supplementary figures, and the somewhat bloated discussion. At this time, I will not concentrate on other more minor considerations.

Reviewer #2 (Remarks to the Author):

This is a very interesting manuscript building on previous observation from the group on P2Y11 in hyperglycemic myocytes. The work utilizes state of the art techniques and presents the data in a transparent manner. The key significance of the data comes from understanding how a molecular complex in myocytes is organized and function in pathological conditions such as hyperglycemia.

The PLAs in Fig 5 are extensive, and do show disruption when AKAP5 KO cells are used. There are some additional controls here that would provide rigor. These include first the use of IgG. In addition, this reviewer would like to see another AKAP-isoform knockout did NOT have an effect (e.g., AKAP150) to demonstrate specificity of the AKAP5 interaction. Along those same lines, what happens to the complex when NF546 is added? Is it always in complex?

It would be helpful to visualize the complex at the membrane of myocytes. This could be done by immunogold or high resolution microscopy.

There are reports of sex differences between males and females with L-type channels in terms of activation and function. This reviewer does not suggest an entire study on the subject, but it would be important to note in the manuscript with data whether the association observed here is

consistent between sexes.

P2Y11 is an interesting purinergic receptor, and until recently didn't really have a mouse gene orthologue (only human). It remains fairly controversial and some have suggested there is no equivalent in mouse (e.g., see review by Dresisig and Kornum, *Purinergic Sign*, 2016). However, this group has done significant work to provide evidence to the contrary (pub in eLife). The use of NF546 mollifies many of the concerns in this manuscript. Regardless, the P2Y11 receptor has been shown to have adenylyl cyclase associations, but also phosphoinositide. Has the phosphoinositide component been checked to be a part of the proposed complex?

Again in relation to P2Y11, many reports indicate an association between P2Y1 and P2Y11 (e.g., Hoffman et al, *JBC* 2008). What is the evidence P2Y11 is acting independently in this complex? What happens to the myogenic tone when MRS 2365 is added?

Figure 4A myogenic tone, the AKAP5 knockouts should be statistically compared against the wild type.

For the proposed mechanism in Figure 6C, how do the nucleotides leave the cell? This is not implying a need for experiments, but what is the evidence in the literature for ATP release from smooth muscle cells?

Minor:

For clarification, were "wild type" mice littermates for AC5^{-/-} and AKAP5^{-/-}? If so, please state that and the rationale for why they were/were not used.

Reviewer #3 (Remarks to the Author):

The study by Paz Prada et al. presents new information on the role of AKAP5 as a central protein in a multiprotein complex including P2Y11, AC5, PKA, and CaV1.2, necessary for local cAMP/PKA modulation of L-type Ca²⁺ channels, regulating vascular tone in arterial myocytes in response to glucose.

For the benefit of the general reader, the authors need to do a better job in introducing methodologies and reagents. For example, please describe briefly the Epac1-camps-based FRET biosensor and its purpose, as well as the ht31 peptide, its derivation and purpose. This should be done consistently throughout the text, when applying methods/reagents for the first time. These texts should of course not be lengthy.

The authors use unpassaged myocyte cultures obtained from adipose arteries. They have published on this methodology before and shown that the cultures are "mostly" smooth muscle cells, but they still need to define in this study what "mostly" means. In the Methods section, it seems that a total collagenase digest was used without selection for smooth muscle cells. Describe better please. It is also necessary to show whether the constitutive deletions of either AKAP5 or AC5 changes the expression levels of other genes, in particular the interaction partners. For example, is the decrease in Cav1.2-dependent PLA signals in Fig. 5 a consequence of decreased expression levels?

The in vivo arterial diameter measurements shown in Fig. 4 are well described in the Methods, however, it would be useful to the reader to be shown higher mag images and the line drawn to measure the diameter. Each vessel was measured at one point; how was this point selected? Why was the measurement of vessel diameter not done blinded?

The PLA results are conclusive, but only for in vitro cultures. As PLA can be applied on sections, the authors can test whether complexes exist also in vivo, which would strengthen the conclusions.

Reviewer 1

1) This very exciting paper focuses on the role of the gene product of AKAP5 in G_s -coupled purinergic modulation of $Ca_v1.2$ channels in vascular smooth muscle cells, and its subsequent vasoconstriction. This concept is similar to, and builds on, similar reports for most members of this team on cardiac myocytes and neurons in brain. The physiological process investigated revolves around how extracellular glucose potentiates the activity of vascular $Ca_v1.2$ channels. It has been known for some years that this signaling mechanism involves PKA, AC5 and G_s -coupled $P2Y_{11}$ receptors. The central advance of this manuscript is the orchestration of this signaling pathway by AKAP79/150 by grouping these signaling elements into "nanodomains," a role this

scaffold protein has been shown to play in brain and muscle for over 5 years. The main approaches used to support this hypothesis include pharmacology, AKAP5 KO mice, FRET AKAP5-mediated PKA reporters, vascular tone and flow measurements and PLA assays.

We thank the reviewer for her/his words of excitement and for highlighting the significance and novel aspects of our work.

Based on the reviewer's comment, we have revised the manuscript to further highlight the innovative and significant aspects of our study. As the reviewer points out, the novel and central advancement in this work is the orchestration of an AKAP5-driven nanocomplex formed by G_s-coupled P2Y₁₁/ G_s-coupled P2Y₁₁-like receptor, AC5, PKA and Ca_v1.2 in human and mouse arterial myocytes. Our data then go on to show that formation of this nanocomplex is critical for 1) increased cAMP synthesis, 2) PKA-dependent potentiation of vascular L-type Ca²⁺ channel activity, 3) enhanced vasoconstriction and 4) concomitant changes in blood flow in response to elevated extracellular glucose. This AKAP5-driven nanocomplex may contribute to vascular complications during diabetic hyperglycemia.

To the best of our knowledge, this is the first example of the formation of this AKAP5/G_s-coupled P2Y₁₁/AC5/PKA/Ca_v1.2 nanocomplex and its (patho)physiological relevance in any cell type. Particularly relevant is that this study establishes for the first time an association of AKAP5 with at least the P2Y₁₁ receptor to regulate purinergic signaling and cell function. Given the broad expression of all members of the nanocomplex in different tissues/cells, we believe our results may have wide implications in many fields. We have clarified these points throughout the manuscript.

2) Most of the elements of this signaling pathway have been previously described by about a dozen publications, mostly by this same group, which I don't need to delineate here. Thus, the signaling pathway involved as described here is not novel. What is novel is the role of AKAP5 gene products in clustering these proteins into "nanodomains" in intimate proximity. I believe this hypothesis is likely to be true, however, the data set in this manuscript do not establish it firmly.

Based on the reviewer's comment, we have revised the manuscript to further highlight the innovative and significant aspects of our work. Accordingly, we discovered a novel signaling nanocomplex involving AKAP5 (AKAP79 in humans and AKAP150 in rodents), G_s-coupled P2Y₁₁ receptor, AC5, PKA and L-type Ca_v1.2 channel in human and mouse arterial myocytes. Our data show that this nanocomplex is anchored by AKAP5, and is critical for 1) increased cAMP synthesis, 2) PKA-dependent potentiation of vascular L-type Ca²⁺ channel activity, 3) enhanced vasoconstriction and 4) concomitant changes in blood flow in response to elevated extracellular glucose. This AKAP5-driven nanocomplex may contribute to vascular complications during diabetic hyperglycemia.

To the best of our knowledge, this is the first example of the formation of this nanocomplex and its (patho)physiological relevance in any cell type. Particularly relevant is that this study establishes for the first time an association of AKAP5 with at least the P2Y₁₁ receptor to regulate purinergic signaling and cell function. Given the broad expression of all members of the nanocomplex in different tissues/cells, we believe our results may have wide implications in many fields. We have clarified these points throughout the manuscript.

We have also added additional super-resolution data (as suggested below) to further support our central hypothesis that AKAP5 mediates the assembly of a G_s-coupled P2Y₁₁/AC5/PKA/Ca_v1.2 signaling nanocomplex to trigger localized cAMP synthesis, PKA-dependent potentiation of vascular L-type Ca²⁺ channels and vasoconstriction during diabetic hyperglycemia. These new data are included and discussed in the revised manuscript. We believe these new data firmly establish the formation of the AKAP5/G_s-coupled P2Y₁₁/AC5/PKA/Ca_v1.2 signaling nanocomplex.

3) Although I appreciate the PLA assays as indicative of this hypothesis, as well as the results from AKAP5 KO mice, I think what is lacking are super-resolution imaging of such complexes as has been performed robustly, and well, by members of this team. Indeed, I was surprised that such experiments were not included in this manuscript. I know this team is capable of performing such experiments in an expert way, and I think for this journal, such data should be included. I do not think that the PLA assays presented are sufficient.

To address this question, we performed super-resolution Airyscan confocal imaging and ground state depletion (GSD) super-resolution nanoscopy in the Total Internal Reflection Fluorescence (TIRF) configuration to detect complexes of proteins of interest at the plasma membrane. Our GSD system and approach allows the detection of protein pairs. Using this super-resolution approach, we have determined close clustering/association between subpopulations of Ca_v1.2 and PKA^{1,2}, Ca_v1.2 and AC5³, and Ca_v1.2 and P2Y₁₁². Therefore here, we focused on establishing whether AKAP5 could closely associate with P2Y₁₁, AC5 and Ca_v1.2. We respectfully submit that repeating super-resolution imaging of Ca_v1.2, PKA, AC5 and P2Y₁₁ would not add new information that will alter the conclusions of the current study. This is particularly relevant now that we are in the middle of the COVID-19 pandemic crisis and that access to the lab is restricted, particularly for performing experiments that will not alter conclusions of a study.

Using super-resolution Airyscan microscopy, intensity projection images of arterial myocytes triple labeled for AKAP5/P2Y₁₁/Ca_v1.2 and AKAP5/AC5/Ca_v1.2 and corresponding line profile analysis showed adjacent and/or overlapping fluorescence associated with each of these combinations of proteins (**Figure 5A and 5B**). Subsequent super-resolution GSD reconstruction maps for AKAP5, P2Y₁₁, AC5 and Ca_v1.2 showed that these proteins form cluster of various sizes and density at the plasma membrane of arterial myocytes (**Figure 5C-5E**). Line profile analysis and merged maps of AKAP5 with P2Y₁₁, AC5 or Ca_v1.2 suggest close association between a subset of these proteins (**Figure 5Cii-5Eii**). Histograms of the AKAP5-to-nearest P2Y₁₁, AC5 or Ca_v1.2 distances revealed that the closest centroids of AKAP5-P2Y₁₁, AKAP5-AC5 and AKAP5-Ca_v1.2 were 40 nm, 44 nm and 42 nm, respectively (**Figure 5Ciii-5Eiii**). The percentage of overlap between AKAP5-P2Y₁₁, AKAP5-AC5 and AKAP5-Ca_v1.2 obtained from the experimental reconstruction maps was significantly higher than that observed for a simulated random distribution between these proteins (**Supplementary Figure 6F**). These results suggest close association between subpopulations of AKAP5 with P2Y₁₁, AC5 and Ca_v1.2 in arterial myocytes. Consistent with this, PLA analysis confirmed close association between AKAP5, P2Y₁₁, AC5 and Ca_v1.2 in arterial myocytes (**Figure 6 and Supplementary Figure 7**). Importantly, genetical ablation of AKAP5 prevented/reduced the close association between P2Y₁₁-Ca_v1.2, P2Y₁₁-PKA_{cat}, AC5-PKA_{cat} and Ca_v1.2-PKA_{RIIα}. Altogether, we believe these results provide strong support to our conclusion that pools of AKAP5, P2Y₁₁, AC5, PKA and Ca_v1.2 clusters with each other to form nanomolecular complexes.

4) Minor. I am not fond of the presentation style of the figures. The black bars and traces superimposed on the light pink bars and traces are hard to discern. In addition, there are other issues in the presentation of the data, such as the perhaps superfluous supplementary figures, and the somewhat bloated discussion. At this time, I will not concentrate on other more minor considerations.

We have revised the format of all figures to improve their presentation. We have examined all supplemental figures to make sure that they are necessary to support our primary data and conclusions. After careful consideration, we believe that the inclusion of these data is essential to show transparency and rigor of our approach. Finally, we have revised the Discussion section to avoid any over-interpretation of results and unnecessary statements. By addressing comments from the other two Reviewers, we also expect to have tackled many of the minor considerations from Reviewer 1.

Reviewer 2

1) This is a very interesting manuscript building on previous observation from the group on P2Y₁₁ in hyperglycemic myocytes. The work utilizes state of the art techniques and presents the data in a transparent manner. The key significance of the data comes from understanding how a molecular complex in myocytes is organized and function in pathological conditions such as hyperglycemia.

We thank the reviewer for highlighting the innovative aspects of our work and for recognizing our efforts to present our data in a transparent and rigorous manner.

2) The PLAs in Fig 5 are extensive and do show disruption when AKAP5 KO cells are used. There are some additional controls here that would provide rigor. These include first the use of IgG.

As suggested, we have included this control experiment in the revised version of the manuscript (**Supplementary Figure 9**). Respectfully, we would also like to submit that our group has done perhaps one of the most rigorous validation of the PLA assay in the field of vascular biology. Accordingly, we have validated the technique in multiple additional ways including 1) no primary antibodies, 2) lack of PLA signal between none interacting proteins, and 3) PLA signal when the same protein is probed with different antibodies¹⁻⁴. Thus, we believe the PLA data support our conclusion that pools of AKAP5, P2Y₁₁, AC5 and Ca_v1.2 clusters with each other to form nanomolecular complexes.

3) In addition, this reviewer would like to see another AKAP-isoform knockout did NOT have an effect (e.g., AKAP150) to demonstrate specificity of the AKAP5 interaction.

We would like to clarify that AKAP5 is referred to as AKAP150 in rodents and AKAP79 in humans. Therefore, knocking out AKAP150 will be the same as the experiments already performed in our study with the AKAP5^{-/-} mouse. We have clarified the AKAP5 nomenclature in the revised version of the manuscript to avoid any confusion (**Page 5, Paragraph 2**).

The possibility that other AKAPs could also interact with P2Y₁₁, AC5, PKA and Ca_v1.2 to form nanomolecular complexes that regulate L-type Ca²⁺ channel activity in arterial myocytes is intriguing. Indeed, this possibility is reminiscent of examples in cardiomyocytes in which both AKAP5 and AKAP7 (also referred to as AKAP15 or AKAP18) have been shown to interact with

β 1 adrenergic receptors, AC, PKA and $\text{Ca}_v1.2$ to form macromolecular complexes^{5,6}. However, our data show that the genetic ablation of AKAP5 disrupts key associations between P2Y_{11} , AC5, PKA and $\text{Ca}_v1.2$ with no changes in total protein expression. Functionally, this has the effect of preventing cAMP synthesis, L-type Ca^{2+} current stimulation, vasoconstriction and changes in blood flow in response to elevated extracellular glucose. Moreover, direct activation of P2Y_{11} with NF546 in $\text{AC5}^{-/-}$ cells failed to induce cAMP synthesis, potentiate L-type Ca^{2+} currents and promote vasoconstriction. Altogether, these results suggest a central role for AKAP5 in orchestrating a signaling module formed by P2Y_{11} , AC5, PKA and $\text{Ca}_v1.2$ to regulate cAMP production, L-type Ca^{2+} channel activity, vascular reactivity and blood flow during diabetic hyperglycemia. Thus, whether other AKAPs are necessary for modulation of the aforementioned signaling pathway by glucose is unlikely. Nevertheless, we are currently working to address this possibility in a future comprehensive study. Furthermore, we acknowledge this possibility in the Discussion section of the revised manuscript (see Page 24, Paragraph 2).

4) Along those same lines, what happens to the complex when NF546 is added? Is it always in complex?

A recent study from our group showed that glucose-induced potentiation of L-type Ca^{2+} currents correlated with increased association of AC5 and $\text{Ca}_v1.2$ ³. These data suggest that the complex remains together, and if any, it is strengthened upon activation of the purinergic signaling. In support of this possibility, data in this study show close association between AKAP5, P2Y_{11} , AC5, PKA and $\text{Ca}_v1.2$ in arterial myocytes (Figure 5 and 6). The close functional association between these proteins is necessary for cAMP synthesis, L-type Ca^{2+} channel stimulation and vasoconstriction in response to elevated glucose and application of NF546. Disruption of the complex upon genetic ablation of AKAP5 (or AC5) prevented the glucose and NF546 effects on cAMP synthesis, L-type Ca^{2+} channel activity, vascular reactivity and changes in blood flow. These data provide strong support to the idea that AKAP5, P2Y_{11} , AC5, PKA and $\text{Ca}_v1.2$ are in complex and remain in it upon activation of the purinergic signaling pathway by elevated glucose and NF546. We acknowledge that further examination of whether glucose and NF546 promote the redistribution of AKAP5, P2Y_{11} , AC5, PKA and $\text{Ca}_v1.2$ at the nanoscale level is important, but respectfully submit that these experiments will not alter the conclusions of the current study. This is particularly relevant now that we are in the middle of the COVID-19 pandemic crisis and that access to the lab is restricted, particularly for performing experiments that will not alter conclusions of a study. Nevertheless, we acknowledge this issue in the Discussion section of the revised manuscript (see Pages 20-21).

5) It would be helpful to visualize the complex at the membrane of myocytes. This could be done by immunogold or high resolution microscopy.

To address this question, we performed super-resolution Airyscan confocal imaging and ground state depletion (GSD) super-resolution nanoscopy in the Total Internal Reflection Fluorescence (TIRF) configuration to detect complexes of proteins of interest at the plasma membrane. Our GSD system and approach allows the detection of protein pairs. Using this super-resolution approach, we have determined close clustering/association between subpopulations of $\text{Ca}_v1.2$ and PKA^{1,2}, $\text{Ca}_v1.2$ and AC5³, and $\text{Ca}_v1.2$ and P2Y_{11} ². Therefore here, we focused on establishing whether AKAP5 could closely associate with P2Y_{11} , AC5 and $\text{Ca}_v1.2$. We respectfully submit that repeating super-resolution imaging of $\text{Ca}_v1.2$, PKA, AC5 and P2Y_{11} would not add new information that will alter the conclusions of the current study. This is particularly relevant now that we are in the middle of the COVID-19 pandemic crisis and that access to the lab is restricted, particularly for performing experiments that will not alter conclusions of a study.

Using super-resolution Airyscan microscopy, intensity projection images of arterial myocytes triple labeled for AKAP5/P2Y₁₁/Ca_v1.2 and AKAP5/AC5/Ca_v1.2 and corresponding line profile analysis showed adjacent and/or overlapping fluorescence associated with each of these combinations of proteins (**Figure 5A and 5B**). Subsequent super-resolution GSD reconstruction maps for AKAP5, P2Y₁₁, AC5 and Ca_v1.2 showed that these proteins form cluster of various sizes and density at the plasma membrane of arterial myocytes (**Figure 5C-5E**). Line profile analysis and merged maps of AKAP5 with P2Y₁₁, AC5 or Ca_v1.2 suggest close association between a subset of these proteins (**Figure 5Cii-5Eii**). Histograms of the AKAP5-to-nearest P2Y₁₁, AC5 or Ca_v1.2 distances revealed that the closest centroids of AKAP5-P2Y₁₁, AKAP5-AC5 and AKAP5-Ca_v1.2 were 40 nm, 44 nm and 42 nm, respectively (**Figure 5Ciii-5Eiii**). The percentage of overlap between AKAP5-P2Y₁₁, AKAP5-AC5 and AKAP5-Ca_v1.2 obtained from the experimental reconstruction maps was significantly higher than that observed for a simulated random distribution between these proteins (**Supplementary Figure 6F**). These results suggest close association between subpopulations of AKAP5 with P2Y₁₁, AC5 and Ca_v1.2 in arterial myocytes. Consistent with this, PLA analysis confirmed close association between AKAP5, P2Y₁₁, AC5 and Ca_v1.2 in arterial myocytes (**Figure 6 and Supplementary Figure 7**). Importantly, genetical ablation of AKAP5 prevented/reduced the close association between P2Y₁₁-Ca_v1.2, P2Y₁₁-PKA_{cat}, AC5-PKA_{cat} and Ca_v1.2-PKA_{RIIα}. Altogether, we believe these results provide strong support to our conclusion that pools of AKAP5, P2Y₁₁, AC5, PKA and Ca_v1.2 clusters with each other to form nanomolecular complexes.

6) There are reports of sex differences between males and females with L-type channels in terms of activation and function. This reviewer does not suggest an entire study on the subject, but it would be important to note in the manuscript with data whether the association observed here is consistent between sexes.

We apologize for not clearly stating in the original manuscript that all human data was obtained using arterial myocytes from male and female patients. Our results show that glucose- and NF546-induced cAMP synthesis mediated by an AKAP-PKA complex (**Figure 1**) as well as the formation of a complex involving AKAP5, P2Y₁₁, AC5 and Ca_v1.2 (**Supplementary Figure 7**) are similar in human male and female arterial myocytes. Thus, data were not segregated by sex. Together with prior studies showing close association between Ca_v1.2 and PKA, Ca_v1.2 and P2Y₁₁ and PKA and P2Y₁₁ in human male and female arterial myocytes^{1,2}, results suggest that functional association of the complex is consistent between sexes.

To further address this point, we measured whole-cell barium currents (I_{Ba}) in wild type and AKAP5^{-/-} female cerebral arterial myocytes before and after 20 mM D-glucose (**Supplementary Figure 4**). We found that elevating extracellular glucose from 10 mM to 20 mM induced a significant increase in I_{Ba} in wild type female arterial myocytes. This glucose-induced potentiation of I_{Ba} was not observed in AKAP5^{-/-} female arterial myocytes. These results are similar to those observed in wild type and AKAP5^{-/-} male cerebral arterial myocytes. Thus, results suggest that AKAP5 is necessary for glucose-induced potentiation of L-type Ca²⁺ channels in male and female arterial myocytes. The revised manuscript now includes these new data, clarification on the sex of human arterial myocytes and discussion proposing that functional association of the AKAP5/P2Y₁₁/AC5/PKA/Ca_v1.2 complex is similar between sexes.

7) P2Y₁₁ is an interesting purinergic receptor, and until recently didn't really have a mouse gene orthologue (only human). It remains fairly controversial and some have suggested there is no equivalent in mouse (e.g., see review by Dreisig and Kornum, Purinergic Sign, 2016). However, this group has done significant work to provide

evidence to the contrary (pub in eLife). The use of NF546 mollifies many of the concerns in this manuscript. Regardless, the P2Y₁₁ receptor has been shown to have adenylyl cyclase associations, but also phosphoinositide. Has the phosphoinositide component been checked to be a part of the proposed complex?

We acknowledge and are well aware of the debate surrounding expression of P2Y₁₁ in rodents. We thank the reviewer for acknowledging our efforts to address this issue in a rigorous manner. In our prior study² and here, we have been very careful to use P2Y₁₁-like receptor when referring to data in mice to precisely avoid any confusion as suggested by Dreisig and Kornum⁷ and Kennedy⁸.

We have not explored whether the phosphoinositide component is part of the proposed complex, which are experiments that fall beyond the scope of the current study. We are aware that the P2Y₁₁ receptor couples to G_s and G_q proteins^{7,8}, and that activation of the latter could regulate L-type Ca²⁺ channel function via engagement of the phosphoinositide component, including PKC. Note however, that we have shown that PKA but not PKC activity is necessary for glucose- and NF546-induced potentiation of L-type Ca²⁺ channels and vasoconstriction^{1,2,9}. These results suggest that the phosphoinositide component is likely to play a minimal role, if any, in regulating L-type Ca²⁺ channel activity and vascular reactivity in response to elevated glucose and NF546.

8) Again, in relation to P2Y₁₁, many reports indicate an association between P2Y₁ and P2Y₁₁ (e.g., Hoffman et al, JBC 2008). What is the evidence P2Y₁₁ is acting independently in this complex? What happens to the myogenic tone when MRS 2365 is added?

We have rigorously addressed this issue recently². First, we found that the NF546-induced cAMP synthesis remained intact in cells pretreated with the selective P2Y₁ inhibitor MRS2179 (10 μM). Yet, this NF546-induced cAMP synthesis was completely inhibited in cells exposed to the P2Y₁₁ inhibitor NF340 (10 μM). These results suggest that cAMP production in response to P2Y₁₁ activation is specific to this receptor and does not involve P2Y₁ receptors. Second, myogenic tone was similar under basal conditions and in arteries pretreated with the P2Y₁ inhibitor MRS2179 (10 μM). These results suggest that P2Y₁ does not contribute to regulate basal myogenic tone. Moreover, glucose-induced vasoconstriction was not altered by adding the P2Y₁ inhibitor MRS2179 (10 μM) either before or after elevating extracellular glucose from 10 mM to 20 mM. These results suggest that the P2Y₁ receptor is not involved in glucose-induced vasoconstriction. Altogether, results suggest a key role for P2Y₁₁ receptor function in mediating the elevated glucose and NF546 effects in arterial myocytes that is independent of their potential hetero-oligomerization with P2Y₁ receptors. Because we have rigorously described this issue in our prior study, a brief acknowledgment is provided in the Introduction section of the revised manuscript (see **Pages 4-5**).

9) Figure 4A myogenic tone, the AKAP5 knockouts should be statistically compared against the wild type.

To address this point, we have modified our data presentation to make more direct comparisons of the glucose effects in wild type vs AKAP5^{-/-} and AC5^{-/-} arteries. We have kept the original figures with individual diameter measurements for transparency (see **new Figures 3Aiii, 3Diii and 7B**).

10) For the proposed mechanism in Figure 6C, how do the nucleotides leave the cell? This is not implying a need for experiments, but what is the evidence in the literature for

ATP release from smooth muscle cells?

As requested, we now provide a discussion of possible mechanisms of nucleotide release and other potential autocrine mechanisms in the revised manuscript (see **Pages 24-25**).

Minor:

1) For clarification, were “wild type” mice littermates for AC5^{-/-} and AKAP5^{-/-}? If so, please state that and the rationale for why they were/were not used.

Yes. The AKAP5^{-/-} and AC5^{-/-} mice have been backcrossed into the C57BL/6 background for 10 generations, and age-matched wild type C57BL/6J mice were used as controls. This has been clarified in the revised manuscript (**Page 9 Paragraph 1, Page 11 Paragraph 2, and Methods section**).

Reviewer 3

1) The study by Paz Prada et al. presents new information on the role of AKAP5 as a central protein in a multiprotein complex including P2Y₁₁, AC5, PKA, and Ca_v1.2, necessary for local cAMP/PKA modulation of L-type Ca²⁺ channels, regulating vascular tone in arterial myocytes in response to glucose.

Thank you for highlighting the innovative and significant aspects of our work.

2) For the benefit of the general reader, the authors need to do a better job in introducing methodologies and reagents. For example, please describe briefly the Epac1-camps-based FRET biosensor and its purpose, as well as the ht31 peptide, its derivation and purpose. This should be done consistently throughout the text, when applying methods/reagents for the first time. These texts should of course not be lengthy.

We apologize for not providing a clear description on these topics in the original manuscript. We have revised the manuscript to clarify the use of methodologies and reagents.

3) The authors use unpassaged myocyte cultures obtained from adipose arteries. They have published on this methodology before and shown that the cultures are “mostly” smooth muscle cells, but they still need to define in this study what “mostly” means. In the Methods section, it seems that a total collagenase digest was used without selection for smooth muscle cells. Describe better please.

To address this point, we performed flow cytometry analysis of cultured arterial myocytes with markers for α -smooth muscle actin, fibroblast, endothelial cells and lineage markers. Our results suggest that 89% of unpassaged cultured cells were positive for α -smooth muscle actin. These results are comparable to recent published data by our group³. These data are now included in the revised manuscript (**Supplemental Figure 1B; Page 7, Paragraph 1**). We have also revised the Methods section to further clarify our approach.

4) It is also necessary to show whether the constitutive deletions of either AKAP5 or AC5 changes the expression levels of other genes, in particular the interaction partners. For example, is the decrease in Cav1.2-dependent PLA signals in Fig. 5 a consequence of decreased expression levels?

To address this question, we performed Western blot analysis of P2Y₁₁, AC5, PKA and Ca_v1.2 in wild type and AKAP5^{-/-} arterial lysates. Our data suggest no change in total P2Y₁₁, AC5, PKA and Ca_v1.2 protein expression between wild type and AKAP5^{-/-} arterial lysates. These data are now included in the revised manuscript (**Supplemental Figure 3; Page 9, Paragraph 1**).

5) The in vivo arterial diameter measurements shown in Fig. 4 are well described in the Methods, however, it would be useful to the reader to be shown higher mag images and the line drawn to measure the diameter. Each vessel was measured at one point; how was this point selected? Why was the measurement of vessel diameter not done blinded?

We have revised the manuscript to clarify these issues and improve data presentation. We would like to point out that indeed, the measurements of vessel diameter were performed by undergraduate students that were blind to conditions. This information is now included in the revised manuscript (**Pages 12-13, Figure 4, and Methods section**).

6) The PLA results are conclusive, but only for in vitro cultures. As PLA can be applied on sections, the authors can test whether complexes exist also in vivo, which would strengthen the conclusions.

The PLA assay was performed in freshly dissociated arterial myocytes, and not in cultured smooth muscle cells. Thus, data represents the formation of the AKAP5/P2Y₁₁/AC5/PKA/Ca_v1.2 complex in an *ex vivo* preparation, consistent with reviewer's suggestion. We attempted to perform the PLA assay in tissue sections. Unfortunately, we were not able to obtain reliable data, likely because of the need to optimize the PLA assay for tissue preparation. We are working to perform these experiments in future studies.

To further strengthened our conclusion, we performed super-resolution Airyscan confocal imaging and ground state depletion (GSD) super-resolution nanoscopy in the Total Internal Reflection Fluorescence (TIRF) configuration to detect complexes of proteins of interest at the plasma membrane. Our GSD system and approach allows the detection of protein pairs. Using this super-resolution approach, we have determined close clustering/association between subpopulations of Ca_v1.2 and PKA^{1,2}, Ca_v1.2 and AC5³, and Ca_v1.2 and P2Y₁₁². Therefore here, we focused on establishing whether AKAP5 could closely associate with P2Y₁₁, AC5 and Ca_v1.2. We respectfully submit that repeating super-resolution imaging of Ca_v1.2, PKA, AC5 and P2Y₁₁ would not add new information that will alter the conclusions of the current study. This is particularly relevant now that we are in the middle of the COVID-19 pandemic crisis and that access to the lab is restricted, particularly for performing experiments that will not alter conclusions of a study.

Using super-resolution Airyscan microscopy, intensity projection images of arterial myocytes triple labeled for AKAP5/P2Y₁₁/Ca_v1.2 and AKAP5/AC5/Ca_v1.2 and corresponding line profile analysis showed adjacent and/or overlapping fluorescence associated with each of these combinations of proteins (**Figure 5A and 5B**). Subsequent super-resolution GSD reconstruction maps for AKAP5, P2Y₁₁, AC5 and Ca_v1.2 showed that these proteins form cluster of various sizes and density at the plasma membrane of arterial myocytes (**Figure 5C-5E**). Line profile analysis and merged maps of AKAP5 with P2Y₁₁, AC5 or Ca_v1.2 suggest close association between a subset of these proteins (**Figure 5Cii-5Eii**). Histograms of the AKAP5-to-nearest P2Y₁₁, AC5 or Ca_v1.2 distances revealed that the closest centroids of AKAP5-P2Y₁₁, AKAP5-AC5 and AKAP5-Ca_v1.2 were 40 nm, 44 nm and 42 nm, respectively (**Figure 5Ciii-5Eiii**). The percentage of overlap between AKAP5-P2Y₁₁, AKAP5-AC5 and AKAP5-Ca_v1.2 obtained from

the experimental reconstruction maps was significantly higher than that observed for a simulated random distribution between these proteins (**Supplementary Figure 6F**). These results suggest close association between subpopulations of AKAP5 with P2Y₁₁, AC5 and Ca_v1.2 in arterial myocytes. Consistent with this, PLA analysis confirmed close association between AKAP5, P2Y₁₁, AC5 and Ca_v1.2 in arterial myocytes (**Figure 6 and Supplementary Figure 7**). Importantly, genetical ablation of AKAP5 prevented/reduced the close association between P2Y₁₁-Ca_v1.2, P2Y₁₁-PKA_{cat}, AC5-PKA_{cat} and Ca_v1.2-PKA_{RIIα}. Altogether, we believe these results provide strong support to our conclusion that pools of AKAP5, P2Y₁₁, AC5, PKA and Ca_v1.2 clusters with each other to form nanomolecular complexes.

References

1. Nystoriak, M.A., *et al.* Ser1928 phosphorylation by PKA stimulates the L-type Ca²⁺ channel CaV1.2 and vasoconstriction during acute hyperglycemia and diabetes. *Science signaling* **10**(2017).
2. Prada, M.P., *et al.* A Gs-coupled purinergic receptor boosts Ca(2+) influx and vascular contractility during diabetic hyperglycemia. *eLife* **8**(2019).
3. Syed, A.U., *et al.* Adenylyl cyclase 5-generated cAMP controls cerebral vascular reactivity during diabetic hyperglycemia. *J Clin Invest* **129**, 3140-3152 (2019).
4. Nieves-Cintrón, M., *et al.* Impaired BKCa channel function in native vascular smooth muscle from humans with type 2 diabetes. *Scientific reports* **7**, 14058 (2017).
5. Nichols, C.B., *et al.* Sympathetic stimulation of adult cardiomyocytes requires association of AKAP5 with a subpopulation of L-type calcium channels. *Circ Res* **107**, 747-756 (2010).
6. Jones, B.W., *et al.* Cardiomyocytes from AKAP7 knockout mice respond normally to adrenergic stimulation. *Proc Natl Acad Sci U S A* **109**, 17099-17104 (2012).
7. Dreisig, K. & Kornum, B.R. A critical look at the function of the P2Y₁₁ receptor. *Purinergic Signal* **12**, 427-437 (2016).
8. Kennedy, C. P2Y₁₁ Receptors: Properties, Distribution and Functions. *Advances in experimental medicine and biology* **1051**, 107-122 (2017).
9. Navedo, M.F., Takeda, Y., Nieves-Cintrón, M., Molkenin, J.D. & Santana, L.F. Elevated Ca²⁺ sparklet activity during acute hyperglycemia and diabetes in cerebral arterial smooth muscle cells. *Am J Physiol Cell Physiol* **298**, C211-220 (2010).

REVIEWERS' COMMENTS:

Reviewer #1 (Remarks to the Author):

The manuscript has been substantially revised and nearly all the critiques of the reviewers have been addressed. The considerable explication of the methods used in the analysis (particularly the imaging) and the vastly-improved presentation style of the figures are highly appreciated. In addition, the novelty and "central advance" of the paper have been more clearly highlighted.

Minor. The manuscript needs to be carefully read and a large number of grammatical errors corrected. This is perhaps tedious, but nonetheless, necessary. In some places, the errors can confuse the meaning of the sentence or the clause involved.

Reviewer #2 (Remarks to the Author):

I am satisfied with the authors response to my previous critique.

Reviewer #3 (Remarks to the Author):

The authors have undertaken an ambitious revision. The new Figure 5 is interesting but the text is quite technical and I had some problems appreciating the data.

In Ai, isn't it fair to say that the line intensity profile for P2Y11 doesn't match those for AKAP5 and Cav1.2 very well?

In Ci, Di, Ei, why are we shown two magnified panels for each antibody in the lower row in this panel? is one supposed to see dots that colocalize for e.g. AKAP5 and P2Y11 in the higher mag lower panels? Those are very hard to see by eye and it's not clear how the high mags are useful. Perhaps the authors could work on the legend to Fig 5 to make it a bit more accessible. The B panel is not mentioned.

Reviewer 1

The manuscript has been substantially revised and nearly all the critiques of the reviewers have been addressed. The considerable explication of the methods used in the analysis (particularly the imaging) and the vastly-improved presentation style of the figures are highly appreciated. In addition, the novelty and "central advance" of the paper have been more clearly highlighted.

We thank the reviewer for her/his comments. We are grateful that the reviewer recognized our efforts to provide additional data in support of our hypotheses, and that data presentation and significance are more clearly presented.

Minor. The manuscript needs to be carefully read, and a large number of grammatical errors corrected. This is perhaps tedious, but nonetheless, necessary. In some places, the errors can confuse the meaning of the sentence or the clause involved.

We thank the reviewer for pointing out this issue. We have carefully gone over the manuscript to address this issue.

Reviewer 2

I am satisfied with the authors response to my previous critique.

We thank the reviewer for her/his thorough assessment of our manuscript, and for the helpful comments and suggestions during the initial review period.

Reviewer 3

The authors have undertaken an ambitious revision.

Thank you!

The new Figure 5 is interesting, but the text is quite technical, and I had some problems appreciating the data. In Ai, isn't it fair to say that the line intensity profile for P2Y₁₁ doesn't match those for AKAP5 and Cav1.2 very well?

We have gone over the figure legend to make it less technical and more accessible to a broader audience. Technical parts were moved to the Methods section.

The line intensity profile for P2Y₁₁ in Figure 5Aii has a less tight matching contour compared to line profiles for AKAP5 and Ca_v1.2. This is likely because the P2Y₁₁-associated fluorescence obtained with the super-resolution Airyscan confocal microscope (axial resolution of ~120 nm) shows a more fragmented plasma membrane distribution. Yet, the data show a number of areas where there is a clear coincidence of the three colors, thus suggesting close proximity of a subset of the three proteins. This hypothesis is further supported by the GSD super-resolution data (axial resolution of ~20-40 nm). Together, data indicate close spatial proximity between AKAP5, P2Y₁₁-like receptors, AC5, and Ca_v1.2. We clarify this issue in the revised version of the manuscript (**Page 14, Paragraph 2**).

In Ci, Di, Ei, why are we shown two magnified panels for each antibody in the lower row in this panel? is one supposed to see dots that colocalize for e.g. AKAP5 and P2Y₁₁ in the higher mag lower panels? Those are very hard to see by eye and it's not clear how the high mags are useful.

The two magnified panels under Figures Ci, Di, Ei highlight clusters of the indicated protein in two areas of a cell. These magnified panels are intended to provide a clearer view of cluster size and distribution in two areas of a cell (for comparisons), which are difficult to appreciate in the image of the entire cell (*upper panels*). These magnified panels do not show colocalization between proteins. Distribution between protein pairs can be found in Figures Cii/Ciii, Dii/Diii and Eii/Eiii as well as Supplementary Figure 6. We have revised the manuscript and figure legend to clarify this point (**Page 15, Paragraph 2, and Page 53**).

Perhaps the authors could work on the legend to Fig 5 to make it a bit more accessible. The B panel is not mentioned.

We have gone over the figure legend to make it less technical and more accessible to a broader audience. Technical parts were moved to the Methods section. We now mention the B panel in the revised figure legend.